# Differentially Private Bilevel Optimization: Efficient Algorithms with Near-Optimal Rates

**Andrew Lowy**\*
CISPA Helmholtz Center for Information Security
lowy.andrew1@gmail.com

**Daogao Liu**
Google Research
liudaogao@gmail.com

## Abstract

Bilevel optimization, in which one optimization problem is nested inside another, underlies many machine learning applications with a hierarchical structure—such as meta-learning and hyperparameter optimization. Such applications often involve sensitive training data, raising pressing concerns about individual privacy. Motivated by this, we study differentially private bilevel optimization. We first focus on settings where the outer-level objective is *convex*, and provide novel upper and lower bounds on the excess empirical risk for both pure and approximate differential privacy. These bounds are nearly tight and essentially match the optimal rates for standard single-level differentially private ERM, up to additional terms that capture the intrinsic complexity of the nested bilevel structure. We also provide population loss bounds for bilevel stochastic optimization. The bounds are achieved in polynomial time via efficient implementations of the exponential and regularized exponential mechanisms. A key technical contribution is a new method and analysis of log-concave sampling under inexact function evaluations, which may be of independent interest. In the *non-convex* setting, we develop novel algorithms with state-of-the-art rates for privately finding approximate stationary points. Notably, our bounds do not depend on the dimension of the inner problem.

## 1 Introduction

Bilevel optimization has emerged as a key tool for solving hierarchical learning and decision-making problems across machine learning and beyond. In a bilevel optimization problem, one task (the *upper-level* problem) is constrained by the solution to another optimization problem (the *lower-level* problem). This nested structure arises naturally in a variety of settings, including meta-learning [43], hyperparameter optimization and model selection [21, 29], reinforcement learning [27], adversarial training [49], and game theory [47], where the solution to one problem depends implicitly on the outcome of another. Formally, a bilevel problem can be written as:

$$\min_{x \in \mathcal{X}} \left\{ \Phi(x) := F(x, y^*(x)) \right\} \tag{1}$$
$$\text{s.t. } y^*(x) \in \operatorname{argmin}_{y \in \mathbb{R}^{d_y}} G(x, y),$$

where $x$ and $y$ are the upper- and lower-level variables respectively, $F$ is the upper-level objective, $G$ is the lower-level objective, $\mathcal{X} \subset \mathbb{R}^{d_x}$ is a domain. Solving (1) is challenging due to the dependency of $y^*(x)$ on $x$. The study of algorithms and complexities for solving (1) has received a lot of attention from the optimization and ML communities in recent years [23, 30, 10, 16, 20, 39, 31, 33, 34, 14].

---

\*Authors listed in reverse alphabetical order. Part of this work was completed while the first author was at University of Wisconsin-Madison.

39th Conference on Neural Information Processing Systems (NeurIPS 2025).

In many applications where bilevel optimization can be useful, data privacy is of critical importance. Machine learning models can leak sensitive training data [45, 13, 42]. *Differential privacy* (DP) [18] mitigates this by ensuring negligible dependence on any single data point.

While differentially private optimization has been extensively studied in a variety of settings [9, 6, 4, 8, 22, 35], the community's understanding of DP BLO is limited. Indeed, we are only aware of two prior works on DP BLO [15, 28]. The work of [15] considers *local DP* [26] and does not provide guarantees in the important privacy regime $\varepsilon = O(1)$. On the other hand, [28] provides guarantees for central DP nonconvex BLO with any $\varepsilon > 0$, which we improve over in this work.

In this work, we provide DP algorithms and error bounds for two fundamental BLO problems. The first BLO problem we study is *bilevel empirical risk minimization (ERM)* w.r.t. data set $Z = (z_1, \ldots, z_n) \in \mathcal{Z}^n$:

$$\min_{x \in \mathcal{X}} \left\{ \widehat{\Phi}_Z(x) := \widehat{F}_Z(x, \widehat{y}_Z^*(x)) = \frac{1}{n} \sum_{i=1}^n f(x, \widehat{y}_Z^*(x), z_i) \right\} \qquad \text{(Bilevel ERM)}$$

$$\text{s.t. } \widehat{y}_Z^*(x) = \text{argmin}_{y \in \mathbb{R}^{d_y}} \left\{ \widehat{G}_Z(x, y) = \frac{1}{n} \sum_{i=1}^n g(x, \widehat{y}_Z^*(x), z_i) \right\},$$

where $f : \mathcal{X} \times \mathbb{R}^{d_y} \times \mathcal{Z} \to \mathbb{R}$ and $g : \mathcal{X} \times \mathbb{R}^{d_y} \times \mathcal{Z} \to \mathbb{R}$ are smooth upper- and lower-level loss functions. Second, we consider *bilevel stochastic optimization (SO)*:

$$\min_{x \in \mathcal{X}} \left\{ \Phi(x) := F(x, y^*(x)) = \mathbb{E}_{z \sim P}[f(x, y^*(x), z)] \right\} \qquad \text{(Bilevel SO)}$$

$$\text{s.t. } y^*(x) = \text{argmin}_{y \in \mathbb{R}^{d_y}} \left\{ G(x, y) = \mathbb{E}_{z \sim P}[g(x, y, z)] \right\}.$$

We assume, as is standard, that $g(x, \cdot, z)$ is strongly convex, so $\forall x$ there are unique $\widehat{y}_Z^*(x)$ and $y^*(x)$.

A fundamental open problem in DP BLO is to determine the *minimax optimal error rates* for solving problems Bilevel ERM and Bilevel SO. A natural first step is to consider the convex case:

> **Question 1.** What are the optimal error rates for solving problem Bilevel ERM with DP when $\widehat{\Phi}_Z$ is convex?

Convex $\widehat{\Phi}_Z, \Phi$ arise in a variety of applications [33], including few-shot meta-learning with a shared embedding model [11], biased regularization in hyperparameter optimization [25], fair resource allocation in communication networks [46], and bilevel optimization with smooth convex $f(\cdot, y)$ and quadratic $g(x, \cdot)$ [33].

**Contribution 1.** We give a (nearly) complete answer to **Question 1** for both pure $\varepsilon$-DP and approximate $(\varepsilon, \delta)$-DP, by providing *nearly tight upper and lower bounds*: see Section 3. Our results show that if the smoothness, Lipschitz, and strong convexity parameters are constants, then it is possible to achieve the same rates as standard single-level convex DP-ERM [9], despite the more challenging bilevel setting (e.g., $O(d_x/\varepsilon n)$ for $\varepsilon$-DP bilevel ERM). On the other hand, our lower bound establishes a novel *separation between standard single-level DP optimization and DP BLO*, showing that the error of any algorithm for DP BLO must necessarily depend on the complexity parameters of the lower-level problem (e.g. the Lipschitz parameter of $g(x, \cdot, z)$). Our algorithms are built on the exponential mechanism [40] for $\varepsilon$-DP and the regularized exponential mechanism [24] for $(\varepsilon, \delta)$-DP. We provide *efficient* (i.e. polynomial-time) implementations of these mechanisms for DP BLO and a novel analysis of how function evaluation errors affect log-concave sampling algorithms. Additionally, we provide upper and lower bounds on the excess population risk for DP Bilevel SO.

**DP Nonconvex BLO.** The recent work of [28] provided an $(\varepsilon, \delta)$-DP algorithm $\mathcal{A}$ capable of finding approximate stationary points of nonconvex $\widehat{\Phi}_Z$ such that

$$\mathbb{E}_{\mathcal{A}} \|\nabla \widehat{\Phi}_Z(\mathcal{A}(Z))\| \leq \tilde{O} \left( \left( \frac{\sqrt{d_x}}{\varepsilon n} \right)^{1/2} + \left( \frac{\sqrt{d_y}}{\varepsilon n} \right)^{1/3} \right). \qquad (2)$$

If $d_y$ is large, bound (2) suffers: e.g., if $d_y \geq d_x$, then the bound is $\gtrsim (\sqrt{d_y}/\varepsilon n)^{1/3}$. This leads us to:

> **Question 2.** Can we improve over the state-of-the-art bound in (2) for DP stationary points in nonconvex Bilevel ERM?

**Contribution 2:** We give a positive answer to **Question 2** in Section 4, developing novel DP algorithms that improve over the bound in (2). Our first algorithm $\mathcal{A}_1$ is a simple and efficient second-order DP BLO method that achieves an improved $d_y$-independent bound of

$$\mathbb{E}\|\nabla\widehat{\Phi}_Z(\mathcal{A}_1(Z))\| \leq \tilde{O}\left(\left(\frac{\sqrt{d_x}}{\varepsilon n}\right)^{1/2}\right).$$

Second, we provide an (inefficient) algorithm $\mathcal{A}_2$ that uses the exponential mechanism to "warm start" $\mathcal{A}_1$ using the framework of [37] to obtain a further improved bound in the parameter regime $d_x < n\varepsilon$:

$$\mathbb{E}\|\nabla\widehat{\Phi}_Z(\mathcal{A}_2(Z))\| \leq \tilde{O}\left(\frac{\sqrt{d_x}}{(\varepsilon n)^{3/4}}\right).$$

As detailed in Appendix C.3, our results imply a new state-of-the-art upper bound for DP non-convex bilevel finite-sum optimization:

$$\mathbb{E}\|\nabla\widehat{\Phi}_Z(\mathcal{A}_{\mathrm{DP}}(Z))\| \leq \tilde{O}\left(\left(\frac{\sqrt{d_x}}{\varepsilon n}\right)^{1/2} \wedge \frac{\sqrt{d_x}}{(\varepsilon n)^{3/4}} \wedge \frac{d_x}{\varepsilon n} \wedge 1\right). \tag{3}$$

## 1.1 Technical overview

We develop and utilize several novel algorithmic and analytic techniques to obtain our results.

**Techniques for convex DP BLO:** Our algorithms are built on the exponential and regularized exponential mechanisms [40, 24]. A key challenge is to implement these algorithms efficiently in BLO, where one lacks access to $\widehat{y}_Z^*(x)$ and hence cannot directly query $\widehat{\Phi}_Z(x)$. To overcome this challenge, we provide a novel analysis of log-concave sampling with inexact function evaluations, building on the grid-walk algorithm of [3] and the approach of [9]. To do so, we prove a bound on the conductance of the *perturbed* Markov chain arising from the grid-walk with perturbed/inexact function evaluation, as well as a bound on the relative distance between the original and perturbed stationary distributions. We believe these techniques and analyses may be of independent interest, since there are many problems beyond BLO where access to exact function evaluations is unavailable.

To prove our lower bounds, we construct a novel bilevel hard instance with linear upper-level $f$ and quadratic lower-level $g$. This allows us to chain together the $x$ and $y$ variables, control $\widehat{y}_Z^*(x)$, and reduce BLO to mean estimation. By carefully scaling our hard instance, we obtain our lower bound.

**Techniques for nonconvex DP BLO:** In the nonconvex setting, our algorithm uses a second-order approximation $\overline{\nabla}\widehat{F}_Z(x_t, y_{t+1}) \approx \nabla\widehat{\Phi}_Z(x_t)$ in order to approximate gradient descent run on $\widehat{\Phi}_Z$. A key insight is that we can obtain a better bound by getting a high-accuracy *non-private* approximate solution $y_{t+1} \approx \widehat{y}_Z^*(x_t)$ and then noising $\overline{\nabla}\widehat{F}_Z(x_t, y_{t+1})$, rather than privatizing $y_{t+1}$. To prove such an approach can be made DP, we require a careful sensitivity analysis that leverages perturbation inequalities from numerical analysis. Further, we build a two-step algorithm on our novel second-order algorithm by leveraging the warm-start framework of [38].

## 2 Preliminaries

**Notation and assumptions.** Let $f : \mathcal{X} \times \mathbb{R}^{d_y} \times \mathcal{Z} \to \mathbb{R}$ and $g : \mathcal{X} \times \mathbb{R}^{d_y} \times \mathcal{Z} \to \mathbb{R}$ be loss functions, with $\mathcal{X} \subset \mathbb{R}^{d_x}$ being a closed convex set of $\ell_2$-diameter $D_x \in [0, \infty]$. The data universe $\mathcal{Z}$ can be any set. $P$ denotes any data distribution on $\mathcal{X}$. Let $\|\cdot\|$ denote the $\ell_2$ norm when applied to vectors. When applied to matrix $A$, $\|A\| := s_{\max}(A) = \sqrt{\lambda_{\max}(AA^T)}$ denotes the $\ell_2$ operator norm, which is the largest singular value of $A$. Function $h : \mathcal{X} \to \mathbb{R}$ is *L-Lipschitz* if $|h(x) - h(x')| \leq L\|x - x'\|$ for all $x, x' \in \mathcal{X}$. Function $h : \mathcal{X} \to \mathbb{R}$ is *$\mu$-strongly convex* if $h(\alpha x + (1-\alpha)x') \leq \alpha h(x) + (1-\alpha)h(x') - \frac{\alpha(1-\alpha)\mu}{2}\|x - x'\|^2$ for all $\alpha \in [0, 1]$ and all $x, x' \in \mathcal{X}$. If $\mu = 0$, we say $h$ is *convex*. The *excess (population) risk* of a randomized algorithm $\mathcal{A}$ with

output $\widehat{x} = \mathcal{A}(Z)$ on loss function $h(x, z)$ is $\mathbb{E}_{\mathcal{A}, Z}[H(\widehat{x})] - H^*$, where $H(x) = \mathbb{E}_{z \sim P}[h(x, z)]$ and $H^* := \inf_x H(x)$. If $\widehat{H}_Z(x) = \frac{1}{n} \sum_{i=1}^n h(x, z_i)$ is an empirical loss function w.r.t. data set $Z$, then the *excess empirical risk* of $\mathcal{A}$ is $\mathbb{E}_{\mathcal{A}}[\widehat{H}_Z(\widehat{x})] - \widehat{H}^*$. Denote $a \wedge b := \min(a, b)$. For functions $\varphi$ and $\psi$ of input parameters $\theta$, we write $\varphi \lesssim \psi$ if there is an absolute constant $C > 0$ such that $\varphi(\theta) \le C\psi(\theta)$ for all permissible values of $\theta$. We use $\widetilde{O}$ to hide logarithmic factors. Denote by $\nabla J(x, y(x), z) = \nabla_x J(x, y(x), z) + \nabla y(x)^T \nabla_y J(x, y(x), z)$ the gradient of function $J$ w.r.t. $x$.

We assume, as is standard in DP optimization, that the loss functions are Lipschitz continuous, and that $g(x, \cdot, z)$ is strongly convex—a standard assumption in the BLO literature:

**Assumption 2.1.**      *1. $f(\cdot, y, z)$ is $L_{f,x}$-Lipschitz in $x$ for all $y, z$.*

     *2. $f(x, \cdot, z)$ is $L_{f,y}$-Lipschitz in $y$ for all $x, z$.*

     *3. $g(x, \cdot, z)$ is $\mu_g$-strongly convex in $y$.*

     *4. There exists a compact set $\mathcal{Y} \subset \mathbb{R}^{d_y}$ with $\{\widehat{y}_Z^*(x)\}_{x \in \mathcal{X}} \subseteq \mathcal{Y}$ for ERM or $\{y^*(x)\}_{x \in \mathcal{X}} \subseteq \mathcal{Y}$ for SO such that $g(x, \cdot, z)$ is $L_{g,y}$-Lipschitz on $\mathcal{Y}$.*

Note that $D_y := \text{diam}(\mathcal{Y}) \le \frac{L_{g,y}}{\mu_g}$ by Assumption 2.1. Some of our algorithms additionally require:

**Assumption 2.2.** *For all $x, x'y, y', z$ we have:*

     *1. $\|\nabla_y f(x, y, z) - \nabla_y f(x, y', z)\| \le \beta_{f,yy}\|y - y'\|$.*

     *2. $\|\nabla_x f(x, y, z) - \nabla_x f(x', y, z)\| \le \beta_{f,xx}\|x - x'\|$.*

     *3. $\|\nabla_x f(x, y, z) - \nabla_x f(x, y', z)\| \le \beta_{f,xy}\|y - y'\|$ and $\|\nabla_y f(x, y, z) - \nabla_y f(x', y, z)\| \le \beta_{f,xy}\|x - x'\|$.*

     *4. $\|\nabla_{xy}^2 g(x, y, z)\| \le \beta_{g,xy}$ and $\|\nabla_{yx}^2 g(x, y, z)\| \le \beta_{g,xy}$.*

     *5. $\|\nabla_{yy}^2 g(x, y, z)\| \le \beta_{g,yy}$.*

     *6. $\|\nabla_{xy}^2 g(x, y, z) - \nabla_{xy}^2 g(x', y, z)\| \le M_{g,xy}\|x - x'\|$, $\|\nabla_{yx}^2 g(x, y, z) - \nabla_{yx}^2 g(x', y, z)\| \le M_{g,xy}\|x - x'\|$, and $\|\nabla_{yy}^2 g(x, y, z) - \nabla_{yy}^2 g(x', y, z)\| \le M_{g,yy}\|x - x'\|$.*

     *7. $\|\nabla_{xy}^2 g(x, y, z) - \nabla_{xy}^2 g(x, y', z)\| \le C_{g,xy}\|y - y'\|$, $\|\nabla_{yx}^2 g(x, y, z) - \nabla_{yx}^2 g(x, y', z)\| \le C_{g,xy}\|y - y'\|$, and $\|\nabla_{yy}^2 g(x, y, z) - \nabla_{yy}^2 g(x, y', z)\| \le C_{g,yy}\|y - y'\|$.*

Assumption 2.2 is standard for second-order optimization methods and is essentially the same as the [28, Assumptions 2.5 and 2.6], but we define the different smoothness parameters at a more granular level to get more precise bounds. As discussed in [23], these assumptions are satisfied in important applications of BLO, such as model selection and hyperparameter tuning with logistic loss (or another loss with bounded gradient and Hessian) and some Stackelberg game models.

**Differential Privacy.**      Differential privacy prevents any adversary from inferring much more about any individual's data than if that data had not been used for training.

**Definition 2.3** (Differential Privacy). Let $\varepsilon \ge 0$, $\delta \in [0, 1)$. Randomized algorithm $\mathcal{A} : \mathcal{Z}^n \to \mathcal{W}$ is $(\varepsilon, \delta)$-*differentially private* (DP) if for any two datasets $Z = (z_1, \ldots, z_n)$ and $Z' = (z_1', \ldots, z_n')$ that differ in one data point (i.e. $z_i \ne z_i'$, $z_j = z_j'$ for $j \ne i$) and any measurable set $S \subset \mathcal{X}$, we have

$$\mathbb{P}(\mathcal{A}(Z) \in S) \le e^\varepsilon \mathbb{P}(\mathcal{A}(Z') \in S) + \delta.$$

Algorithmic preliminaries on DP are given in Appendix A.

## 3   Private convex bilevel optimization

In this section, we characterize the optimal excess risk bounds for DP convex bilevel ERM:

**Theorem 3.1** (Convex DP BLO - Informal). *Let $\widehat{\Phi}_Z$ and $\Phi$ be convex ($\forall Z \in \mathcal{Z}^n$) and grant Assumption 2.1. Then, there is an efficient $\varepsilon$-DP algorithm with output $\widehat{x}$ such that*

$$\mathbb{E}\widehat{\Phi}_Z(\widehat{x}) - \widehat{\Phi}_Z^* \leq \tilde{O}\left(\frac{d_x}{\varepsilon n}\right).$$

*If Assumption 2.2 parts 3-4 hold, then there is an efficient $(\varepsilon, \delta)$-DP algorithm with output $\widehat{x}$ s.t.*

$$\mathbb{E}\widehat{\Phi}_Z(\widehat{x}) - \widehat{\Phi}_Z^* \leq O\left(\frac{\sqrt{d_x \log(1/\delta)}}{\varepsilon n}\right)$$

*Moreover, the above upper bounds are tight (optimal) up to logarithmic factors.*

The following subsections contain formal statements capturing the precise dependence on the problem parameters given in Assumptions 2.1 and 2.2 and runtime bounds. We also provide bounds for DP convex bilevel SO.

### 3.1 Conceptual algorithms and excess risk upper bounds

This section contains our conceptual algorithms (ignoring efficiency considerations) and precise excess risk upper bounds.

**Pure $\varepsilon$-DP.** Consider the following sampler for DP bilevel ERM, which is an instantiation of the *exponential mechanism* [40]: Given $Z \in \mathcal{Z}^n$, sample $\widehat{x} = \widehat{x}(Z) \in \mathcal{X}$ with probability

$$\propto \exp\left(-\frac{\varepsilon}{2s}\widehat{\Phi}_Z(\widehat{x})\right), \quad \text{where} \tag{4}$$

$$s := \frac{2}{n}\left[L_{f,x}D_x + L_{f,y}D_y\right] + \frac{4L_{f,y}L_{g,y}}{\mu_g}.$$

**Theorem 3.2.** *Grant Assumption 2.1 and suppose $\widehat{\Phi}_Z$ is convex. The Algorithm in (4) is $\varepsilon$-DP and*

$$\mathbb{E}[\widehat{\Phi}_Z(\widehat{x}) - \widehat{\Phi}_Z^*] \leq O\left(\frac{d_x}{\varepsilon n}\left[L_{f,x}D_x + L_{f,y}D_y + \frac{L_{f,y}L_{g,y}}{\mu_g}\right]\right).$$

We defer the proof to Appendix B and describe the efficient implementation in Section 3.2. If $\Phi$ is not convex, then privacy still holds and the same excess risk holds up to logarithmic factors. The key step in the privacy proof is to upper bound the sensitivity of the score function $\widehat{\Phi}_Z(x)$ by $s$, by leveraging Assumption 2.1 and the fact that $\|\widehat{y}_Z^*(x) - \widehat{y}_{Z'}^*(x)\| \leq 2L_{g,y}/\mu_g n$ for adjacent $Z \sim Z'$.

**Approximate $(\varepsilon, \delta)$-DP.** Consider the following instantiation of the *regularized exponential mechanism* [24]: Given $Z$, sample $\widehat{x} = \widehat{x}(Z)$ from probability density function

$$\propto \exp(-k(\widehat{\Phi}_Z(\widehat{x}) + \mu\|\widehat{x}\|^2)), \quad \text{where} \tag{5}$$

$$k = O\left(\frac{\mu n^2 \epsilon^2}{G^2 \log(1/\delta)}\right) \quad \text{and}$$

$$G = L_{f,x} + \frac{L_{f,y}\beta_{g,xy}}{\mu_g} + \frac{L_{g,y}\beta_{f,xy}}{\mu_g},$$

where $\mu$ is an algorithmic parameter that we will assign (not to be confused with $\mu_g$).

**Theorem 3.3** (Informal). *Grant Assumption 2.1 and parts 3 and 4 of Assumption 2.2. Assume $\widehat{\Phi}_Z$ and $\Phi$ are convex for all $Z \in \mathcal{Z}^n$. There exists a choice of $\mu$ and $k$ such that Algorithm (5) is $(\varepsilon, \delta)$-DP and achieves excess empirical risk*

$$\mathbb{E}\widehat{\Phi}_Z(\widehat{x}) - \widehat{\Phi}_Z^* \leq O\left(\left(L_{f,x} + \frac{L_{f,y}\beta_{g,xy}}{\mu_g} + \frac{L_{g,y}\beta_{f,xy}}{\mu_g}\right)D_x\frac{\sqrt{d_x \log(1/\delta)}}{\varepsilon n}\right).$$

*Further, if $Z \sim P^n$ are independent samples, the excess population risk with a different choice of $k, \mu$ is*

$$\mathbb{E}\Phi(\widehat{x}) - \Phi^* \leq O\left(\frac{\sqrt{d_x \log(1/\delta)}}{\varepsilon n} + \frac{1}{n^{1/4}}\right).$$

Refer to Theorem B.8 in Appendix B for the precise population loss bound with proper units. The main idea of the privacy proof (in Appendix B) is to show that $\widehat{\Phi}_Z - \widehat{\Phi}_{Z'}$ is $2\left(\frac{L_{f,x}}{n} + \frac{L_{f,y}\beta_{g,xy}}{\mu_g n} + \frac{L_{g,y}\beta_{f,xy}}{n\mu_g}\right)$-Lipschitz and then compare the *privacy curve* [5] between the distributions $Q$ and $Q'$ (corresponding to (5) with data $Z$ and $Z'$ respectively) to the privacy curve between two Gaussians, by leveraging [24, Theorem 4.1].

Obtaining our dimension-independent generalization error bound involves a fairly long "ghost sample" argument that leverages the Wasserstein distance bound for log-concave distributions, Kantorovich-Rubinstein duality, and the Efron-Stein inequality. One can also obtain an $O(\sqrt{d/n})$ generalization error bound by a uniform convergence argument; we omit those details here.

*Remark* 3.4 (Near-optimality for ERM). The bounds in Theorem 3.2 and 3.3 nearly match the optimal bounds for standard single-level DP ERM [9], e.g. $\Theta(L_{f,x}D_x\sqrt{d_x \log(1/\delta)}/\varepsilon n)$ for $(\varepsilon, \delta)$-DP ERM [9, 48], except for the addition of two terms capturing the complexity of the bilevel problem: For $\varepsilon$-DP ERM, the additional terms are $O(L_{f,y}D_y d_x/\varepsilon n)$ and $O((L_{f,y}L_{g,y}/\mu_g)d_x/\varepsilon n)$. Our lower bound in Theorem 3.10 shows that the first additional term is necessary. We conjecture that the second additional term is also necessary and that our upper bound is *tight up to an absolute constant*. This conjecture is clearly true in the parameter regime $L_{g,y}/\mu_g \approx D_y$. For $(\varepsilon, \delta)$-DP, the additional terms scale with $O((L_{f,y}\beta_{g,xy}/\mu_g + L_{g,y}\beta_{f,xy}/\mu_g)D_x)$. Our lower bound in Theorem 3.10 shows that dependence on $L_{f,y}$ is necessary and that the $L_{f,y}\beta_{g,xy}/\mu_g$ term is tight in the parameter regime $D_y \approx D_x\beta_{g,xy}/\mu_g$. If also $D_x\beta_{g,xy}/ \lesssim L_{g,y}$, then the bounds in Theorem 3.3 are tight up to an absolute constant factor.

*Remark* 3.5 (Suboptimality for SO). There is a gap between our population risk upper bound in Theorem 3.3 and the lower bounds in [6] and Remark 3.11 for single-level SCO and bilevel SO respectively. We conjecture that our lower bound in Remark 3.11 is nearly tight and that our upper bound is suboptimal. We leave it as future work to investigate this conjecture.

## 3.2 Efficient implementation of conceptual algorithms

In many practical applications of optimization and sampling algorithms, we face unavoidable approximation errors when evaluating functions. Given any $x$, we may not get the exact $\widehat{y}_Z^*(x)$ in solving the low-level optimization, which means we may introduce a small error each time we compute the function value of $f(x, \widehat{y}_Z^*(x), z)$. This section analyzes how such small function evaluation errors affect log-concave sampling algorithms. We establish bounds on the impact of errors bounded by $\zeta$ on the conductance, mixing time, and distributional accuracy of Markov chains used for sampling. We then develop an efficient implementation based on the [9] approach that maintains polynomial time complexity while providing formal guarantees on sampling accuracy in the presence of function evaluation errors. As a corollary of our developments, we obtain Theorem 3.1.

Our approach builds on the classic Grid-Walk algorithm of [3] for sampling from log-concave distributions. Let $F(\cdot)$ be a real positive-valued function defined on a cube $A = [a, b]^d$ in $\mathbb{R}^d$. Let $f(\theta) = -\log F(\theta)$ and suppose there exist real numbers $\alpha, \beta$ such that:

$$|f(x) - f(y)| \leq \alpha \left( \max_{i \in [1,d]} |x_i - y_i| \right),$$

$$f(\lambda x + (1-\lambda)y) \geq \lambda f(x) + (1-\lambda)f(y) - \beta,$$

for all $x, y \in A$ and $\lambda \in [0, 1]$. The algorithm of [3], detailed in Appendix B.3.1 for completeness, samples from a distribution $\nu$ on the continuous domain $A$ such that for all $\theta \in A$, $|\nu(\theta) - cF(\theta)| \leq \zeta$, where $c$ is a normalization constant and $\zeta > 0$. The algorithm defines a random walk (which is a Markov Chain) on the centers of small subcubes that partition $A$ and form the state space $\Omega \subset A$. The final output of the algorithm is a point $x \in A$, returned with probability close to $F(x)$.

Next, we briefly outline our analysis how the Grid-Walk algorithm behaves when the function $F$ can only be evaluated with some bounded error, resulting in a "perturbed" Markov chain.

**Conductance bound with function evaluation errors.** For a Markov chain with state space $\Omega$, transition matrix $P$ and stationary distribution $q$, its conductance $\varphi$ measures how well the chain mixes, i.e. how quickly it converges to its stationary distribution:

$$\varphi := \min_{S \subset \Omega : 0 < q(S) \leq 1/2} \frac{\sum_{x \in S, y \in \Omega \setminus S} q(x)P_{xy}}{q(S)}.$$

We analyze how function evaluation errors affect Grid-Walk conductance:

**Lemma 3.6** (Conductance with Function Evaluation Errors). *Let $P$ be the transition matrix of the original Markov chain in the grid-walk algorithm of Section B.3.1 based on function $f$, with state space $\Omega$ and conductance $\varphi$. Let $P'$ be the transition matrix of the perturbed chain based on $f'$ where $f'(\theta) = f(\theta) + \zeta(\theta)$ with $|\zeta(\theta)| \leq \zeta$ for all $\theta \in \Omega$, where $\zeta(\cdot)$ is a bounded error function. Then the conductance $\varphi'$ of the perturbed chain satisfies:*

$$\varphi' \geq e^{-6\zeta} \varphi.$$

**Relative distance bound between $F$ and $F'$.** We now analyze how function evaluation errors affect the distributional distance between the original and perturbed stationary distributions.

**Lemma 3.7** (Distance Between $F$ and $F'$). *Let $F(\theta) = e^{-f(\theta)}$ and $F'(\theta) = e^{-f'(\theta)}$ where $f'(\theta) = f(\theta) + \zeta(\theta)$ with $|\zeta(\theta)| \leq \zeta$ for all $\theta \in A$. Then,*

$$e^{-\zeta} \leq \frac{F'(\theta)}{F(\theta)} \leq e^{\zeta}, \quad \forall \theta \in A.$$

*Furthermore, if we define the distributions $\pi(\theta) \propto F(\theta)$ and $\pi'(\theta) \propto F'(\theta)$, then:*

$$\mathrm{Dist}_\infty(\pi', \pi) := \sup_{\theta \in A} \left| \log \frac{\pi'(\theta)}{\pi(\theta)} \right| \leq 2\zeta.$$

**Mixing time analysis.** For a Markov chain with state space $\Omega$, transition matrix $P$, and stationary distribution $\pi$, the *mixing time* $t_{\mathrm{mix}}(\epsilon)$ with respect to the $L_\infty$-distance is defined as:

$$t_{\mathrm{mix}}(\epsilon) := \min\{t \geq 0 : \max_{x \in \Omega} \mathrm{Dist}_\infty(P^t(x, \cdot), \pi(\cdot)) \leq \epsilon\}, \tag{6}$$

for any $\epsilon \geq 0$. We determine the number of steps required for $L_\infty$ convergence with perturbed $F$:

**Lemma 3.8** (Impact on Mixing Time). *The mixing time $t'_{mix}(\epsilon)$ of the perturbed chain to achieve $L_\infty$-distance $\epsilon$ to its stationary distribution satisfies:*

$$t'_{mix}(\epsilon) \leq e^{12\zeta} \cdot O\left( \frac{\alpha^2 \tau^2 d^2}{\epsilon^2} e^\epsilon \max\left\{ d \log \frac{\alpha \tau \sqrt{d}}{\epsilon}, \alpha \tau \right\} \right).$$

**Efficient implementation.** Leveraging our analysis of how function evaluation errors affect conductance, mixing time, and distributional distance, we develop an efficient algorithm for sampling from log-concave distributions in the presence of such errors. Our approach builds upon the framework developed by [9], extending it to handle approximation errors.

**Theorem 3.9** (Log-Concave Sampling with Function Evaluation Error). *Let $C \subset \mathbb{R}^d$ be a convex set and $f : C \to \mathbb{R}$ be a convex, $L$-Lipschitz function. Suppose we have access to an approximate function evaluator that returns $f'(\theta) = f(\theta) + \zeta(\theta)$ where $|\zeta(\theta)| \leq \zeta$ for all $\theta \in C$, and $\zeta = O(1)$ is a constant independent of dimension. There exists an efficient algorithm that outputs a sample $\theta \in C$ from a distribution $\mu'$ such that:*

$$\mathrm{Dist}_\infty(\mu', \pi) \leq 2\zeta + \xi \tag{7}$$

*where $\pi(\theta) \propto e^{-f(\theta)}$ is the target log-concave distribution and $\delta > 0$ is an arbitrarily small constant. This algorithm runs in time $O(e^{12\zeta} \cdot d^3 \cdot poly(L, \|C\|_2, 1/\xi))$.*

The efficiency claims in Theorem 3.1 follow as corollaries of Theorem 3.9: see Appendix B.3.

## 3.3 Excess risk lower bounds

If the problem parameters (e.g., Lipschitz, smoothness) are constants, then the upper bounds in Theorems 3.2 and 3.3 are tight and match known lower bounds for standard single-level DP ERM and SCO [9, 6]. In this section, we go a step further and provide novel lower bounds illustrating that the dependence of our bounds on $L_{f,y} D_y$ (or a quantity larger than this) is necessary, thereby establishing a *novel separation between single-level DP optimization and DP BLO*:

**Theorem 3.10** (Excess risk lower bounds for DP ERM). *1. Let $\mathcal{A}$ be $\varepsilon$-DP. Then, there exists a data set $Z \in \mathcal{Z}^n$ and a convex bilevel ERM problem instance satisfying Assumptions 2.1 and 2.2 with $\mu_g = \Theta(L_{g,y}/D_y)$ such that*

$$\mathbb{E}\widehat{\Phi}_Z(\mathcal{A}(Z)) - \widehat{\Phi}_Z^* = \Omega\left((L_{f,x}D_x + L_{f,y}D_y)\min\left\{1, \frac{d_x}{n\varepsilon}\right\}\right).$$

*2. Let $\mathcal{A}$ be $(\varepsilon, \delta)$-DP with $2^{-\Omega(n)} \leq \delta \leq 1/n^{1+\Omega(1)}$. Then, there exists a data set $Z \in \mathcal{Z}^n$ and a convex bilevel ERM problem instance satisfying Assumptions 2.1 and 2.2 with $\mu_g = \Theta(L_{g,y}/D_y)$ such that*

$$\mathbb{E}\widehat{\Phi}_Z(\mathcal{A}(Z)) - \widehat{\Phi}_Z^* = \Omega\left((L_{f,x}D_x + L_{f,y}D_y)\min\left\{1, \frac{\sqrt{d_x\log(1/\delta)}}{n\varepsilon}\right\}\right).$$

By comparing the lower bounds in Theorem 3.10 with the bounds in [9], one sees that the *DP bilevel ERM is harder (in terms of minimax error) than standard single-level DP ERM if $L_{f,y}D_y > L_{f,x}D_x$.*

See Appendix B.4 for the proof. A key challenge is in constructing the right $f$ and $g$ to chain together the $x$ and $y$ variables and obtain the desired $L_{f,y}D_y$ scaling term.

*Remark* 3.11 (Bilevel SO lower bounds). One can obtain lower bounds on the excess population risk that are larger than the excess empirical risk bounds in Theorem 3.10 by an additive $L_{f,x}D_x(1/\sqrt{n})$, via the reduction in [7].

## 4 Private non-convex bilevel optimization

In this section, we provide novel algorithms with state-of-the-art guarantees for privately finding approximate stationary points of non-convex $\widehat{\Phi}_Z$ (see (3)).

### 4.1 An iterative second-order method

Assume for simplicity that $\mathcal{X} = \mathbb{R}^{d_x}$ so that the optimization problem is unconstrained.[2] A natural approach to solving Bilevel ERM is to use a gradient descent scheme, where we iterate

$$x_{t+1} = x_t - \eta\nabla\widehat{\Phi}_Z(x_t). \tag{8}$$

By the implicit function theorem, we have (c.f. [23]):

$$\nabla\widehat{\Phi}_Z(x) = \nabla_x\widehat{F}_Z(x, \widehat{y}_Z^*(x)) - \nabla_{xy}^2\widehat{G}_Z(x, \widehat{y}_Z^*(x))[\nabla_{yy}^2\widehat{G}_Z(x, \widehat{y}_Z^*(x))]^{-1}\nabla_y\widehat{F}_Z(x, \widehat{y}_Z^*(x)).$$

Define the following approximation to $\nabla\widehat{\Phi}_Z(x)$ at $(x, y)$:

$$\overline{\nabla}\widehat{F}_Z(x, y) := \nabla_x\widehat{F}_Z(x, y) - \nabla_{xy}^2\widehat{G}_Z(x, y)[\nabla_{yy}^2\widehat{G}_Z(x, y)]^{-1}\nabla_y\widehat{F}_Z(x, y). \tag{9}$$

Note that $\overline{\nabla}\widehat{F}_Z(x, y) = \nabla\widehat{\Phi}_Z(x)$ if $y = \widehat{y}_Z^*(x)$.

Then to approximate (8) (non-privately), we can iterate (c.f. [23]):

$$y_{t+1} \approx \widehat{y}_Z^*(x_t)$$
$$x_{t+1} = x_t - \eta\overline{\nabla}\widehat{F}_Z(x_t, y_{t+1}). \tag{10}$$

A naive approach to privatizing the iterations (10) is to solve $y_{t+1} \approx \widehat{y}_Z^*(x_t) = \arg\min_y\widehat{G}_Z(x_t, y)$ *privately* at each step (e.g., by running DP-SGD), and then add noise to $\overline{\nabla}\widehat{F}_Z(x_t, y_{t+1})$ before taking a step of noisy GD. (This is similar to how [28] privatized the penalty-based bilevel optimization algorithm of [30].) However, this approach results in a bound $\mathbb{E}\|\nabla\widehat{\Phi}_Z(\widehat{x})\| \leq O(\sqrt{d_x + d_y}/\varepsilon n)^{1/2}$ that depends on $d_y$ due to the bias $\|\overline{\nabla}\widehat{F}_Z(x_t, y_{t+1}) - \nabla\widehat{\Phi}_Z(x_t)\|$ that results from using private $y_{t+1}$. To mitigate this issue and obtain state-of-the-art utility independent of $d_y$, we propose an alternative approach in Algorithm 1: we find an approximate minimizer of $\widehat{G}_Z(x_t, \cdot)$ *non-privately* in line 3.

**Algorithm 1:** A Second-Order DP Bilevel Optimization Algorithm

---
1 **Input:** Dataset $\mathcal{D} = (Z_1, \ldots, Z_n)$, noise scale $\sigma$, initial points $x_0, y_0 \in \mathcal{X} \times \mathcal{Y}$, parameter $\alpha$;
2 **for** $i = 0, \ldots, T - 1$ **do**
3 $\quad$ Find $y_{t+1} \approx \widehat{y}_Z^*(x_t)$ such that $\|y_{t+1} - \widehat{y}_Z^*(x_t)\| \leq \alpha$ (e.g., via SGD or Katyusha [2]);
$\quad\quad$ $x_{t+1} = x_t - \eta \left( \overline{\nabla} \widehat{F}_Z(x_t, y_{t+1}) + u_t \right)$, where $u_t \sim \mathcal{N}(0, \sigma^2 \mathbf{I}_{d_x})$.
4 **end**
5 **Output:** $\widehat{x}_T \sim \mathbf{Unif}(\{x_t\}_{t=1}^T)$.

---

Since $\widehat{G}_Z(x_t, \cdot)$ is a smooth, strongly convex ERM function, we can implement line 3 efficiently using a non-private algorithm such as SGD or Katyusha [2].

Denote $\overline{L} := L_{f,x} + \frac{\beta_{g,xy} L_{f,y}}{\mu_g}$, which is an upper bound on $\|\nabla f(x, \widehat{y}_Z^*(x), z)\|$, and

$$C := \beta_{f,xy} + \frac{\beta_{f,yy} \beta_{g,xy}}{\mu_g} + L_{f,y} \left( \frac{C_{g,xy}}{\mu_g} + \frac{C_{g,yy} \beta_{g,xy}}{\mu_g^2} \right), \tag{11}$$

which satisfies $\|\nabla \widehat{\Phi}_Z(x) - \overline{\nabla} \widehat{F}_Z(x, y)\| \leq C \|\widehat{y}_Z^*(x) - y\|$ for any $x, y$ by [23, Lemma 2.2]. Let

$$K := 2 \left[ \frac{\beta_{f,xy} L_{g,y}}{\mu_g} + 2\overline{L} + \frac{\beta_{g,xy} \beta_{f,yy} L_{g,y}}{\mu_g^2} + \frac{L_{f,y} C_{g,xy} L_{g,y}}{\mu_g^2} + \frac{L_{f,y} \beta_{g,xy} L_{g,y} C_{g,yy}}{\mu_g^3} + \frac{L_{f,y} \beta_{g,yy} \beta_{g,xy}}{\mu_g^2} \right]. \tag{12}$$

**Lemma 4.1** (Sensitivity Bound for Algorithm 1). *For any fixed $x_t$, define the query $q_t : \mathcal{Z}^n \to \mathbb{R}^d$,*

$$q_t(Z) := \overline{\nabla} \widehat{F}_Z(x_t, y_{t+1}),$$

*where $y_{t+1} = y_{t+1}(Z)$ is given in Algorithm 1. If $\alpha \leq \frac{K}{Cn}$ where $C$ and $K$ are defined in Equations (11) and (12), then the $\ell_2$-sensitivity of $q_t$ is upper bounded by $\frac{4K}{n}$.*

The proof of this lemma—in Appendix C—is long. It uses the operator norm perturbation inequality $\|M^{-1} - N^{-1}\| \leq \|M^{-1}\| \|N^{-1}\| \|M - N\|$ to bound the sensitivity of $[\nabla_{yy}^2 \widehat{G}_Z(x_t, y_{y+1})]^{-1}$ in (9).

Now we can state the main result of this subsection:

**Theorem 4.2** (Guarantees of Algorithm 1 for Non-Convex Bilevel ERM - Informal). *Grant Assumptions 2.1 and 2.2. Set $\sigma = 32K \sqrt{T \log(1/\delta)} / n\varepsilon$. Denote the smoothness parameter of $\widehat{\Phi}_Z$ by $\beta_\Phi$, given in Lemma C.2. There are choices of $\alpha, \eta$ s.t. Algorithm 1 is $(\varepsilon, \delta)$-DP and has output satisfying*

$$\mathbb{E} \|\nabla \widehat{\Phi}_Z(\widehat{x}_T)\| \lesssim \left[ K \sqrt{\left( \widehat{\Phi}_Z(x_0) - \widehat{\Phi}_Z^* \right) \beta_\Phi} \frac{\sqrt{d_x \log(1/\delta)}}{\varepsilon n} \right]^{1/2}.$$

The privacy proof leverages Lemma 4.1. Utility is analyzed through the lens of gradient descent with biased, noisy gradient oracle. We choose small $\alpha$ so the bias is negligible and use smoothness of $\widehat{\Phi}_Z$.

## 4.2 "Warm starting" Algorithm 1 with the exponential mechanism

This subsection provides an algorithm that enables an improvement over the utility bound given in Theorem 4.2 in the parameter regime $d_x < n\varepsilon$. Our algorithm is built on the "warm start" framework of [37]: first, we run the exponential mechanism (4) with privacy parameter $\varepsilon/2$ to obtain $x_0$; then, we run $(\varepsilon/2, \delta)$-DP Algorithm 1 with "warm" initial point $x_0$. See Algorithm 2 in Appendix C.2.

**Theorem 4.3** (Guarantees of Algorithm 2 for Non-Convex Bilevel ERM). *Grant Assumptions 2.1 and 2.2. Assume that there is a compact set $\mathcal{X} \subset \mathbb{R}^{d_x}$ of diameter $D_x$ containing an approximate global minimizer $\widehat{x}$ such that $\widehat{\Phi}_Z(\widehat{x}) - \widehat{\Phi}_Z^* \leq \Psi \frac{d}{\varepsilon n}$, where $\Psi := L_{f,x} D_x + L_{f,y} D_y + \frac{L_{f,y} L_{g,y}}{\mu_g}$. Then, there exists an $(\varepsilon, \delta)$-DP instantiation of Algorithm 2 with output satisfying*

$$\mathbb{E} \|\nabla \widehat{\Phi}_Z(x_{priv})\| \leq \tilde{O} \left( \left[ K \Psi^{1/2} \beta_\Phi^{1/2} \right]^{1/2} \left( \frac{d_x \sqrt{\log(1/\delta)}}{(n\varepsilon)^{3/2}} \right)^{1/2} \right).$$

---
[2]Our approach and results readily extend to constrained $\mathcal{X}$ by incorporating proximal steps and measuring utility in terms of the norm of the proximal gradient mapping.

In Appendix C.3, we explain how to deduce the upper bound in (3) by combining Theorems 4.2 and 4.3 with the exponential mechanism using cost function $\|\nabla \widehat{\Phi}_Z(x)\|$.

## 5 Conclusion and discussion

We provided novel algorithms and lower bounds for differentially private bilevel optimization, with near-optimal rates for the convex setting and state-of-the-art rates for the nonconvex setting. There are some interesting open problems for future work to explore: (1) What are the optimal rates for DP *bilevel convex SO*? As discussed in Remark 3.5, we believe that it should be possible, though challenging, to obtain an improved $O(1/\sqrt{n})$ generalization error bound nearly matching the lower bound in Remark 3.11. (2a) What are the *optimal rates for DP nonconvex bilevel ERM and SO*? Since the optimal rates for standard single-level DP nonconvex ERM and SO are still unknown, a first step would be to answer: (2b) *Can we match the SOTA rate for single-level non-convex ERM [38] in BLO*? Incorporating variance-reduction in DP BLO seems challenging. (3) This work was focused on fundamental theoretical questions about DP BLO, but another important direction is to provide *practical implementations and experimental evaluations*.

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

# Appendix

## A  More privacy preliminaries

**Definition A.1** (Sensitivity). Given a function $q : \mathcal{Z}^n \to \mathbb{R}^k$ the $\ell_2$-*sensitivity* of $q$ is defined as

$$\sup_{Z \sim Z'} \|q(Z) - q(Z')\|,$$

where the supremum is taken over all pairs of datasets that differ in one data point.

**Definition A.2** (Gaussian Mechanism). Let $\varepsilon > 0$, $\delta \in (0,1)$. Given a function $q : \mathcal{Z}^n \to \mathbb{R}^k$ with $\ell_2$-sensitivity $\Delta$, the *Gaussian Mechanism* $\mathcal{M}$ is defined by

$$\mathcal{M}(Z) := q(Z) + v$$

where $v \sim \mathcal{N}_k\left(0, \sigma^2 \mathbf{I}_k\right)$ and $\sigma^2 = \frac{2\Delta^2 \log(2/\delta)}{\varepsilon^2}$.

**Lemma A.3** (Privacy of Gaussian Mechanism [19]). *The Gaussian Mechanism is $(\varepsilon, \delta)$-DP.*

If we adaptively query a data set $T$ times, then the privacy guarantees of the $T$-th query is still DP and the privacy parameters degrade gracefully:

**Lemma A.4** (Advanced Composition Theorem [19]). *Let $\varepsilon \geq 0, \delta, \delta' \in [0,1)$. Assume $\mathcal{A}_1, \cdots, \mathcal{A}_T$, with $\mathcal{A}_t : \mathcal{Z}^n \times \mathcal{X} \to \mathcal{X}$, are each $(\varepsilon, \delta)$-DP $\forall t = 1, \cdots, T$. Then, the adaptive composition $\mathcal{A}(Z) := \mathcal{A}_T(Z, \mathcal{A}_{T-1}(Z, \mathcal{A}_{T-2}(X, \cdots)))$ is $(\varepsilon', T\delta + \delta')$-DP for*

$$\varepsilon' = \sqrt{2T \ln(1/\delta')}\varepsilon + T\varepsilon(e^\varepsilon - 1).$$

## B  Proofs for Section 3.1

### B.1  Conceptual algorithms and excess risk upper bounds

**Pure $\varepsilon$-DP.**  We restate and prove the guarantees of the $\varepsilon$-DP exponential mechanism for BLO below:

**Theorem B.1** (Re-statement of Theorem 3.2). *Grant Assumption 2.1 and suppose $\widehat{\Phi}_Z$ is convex. The Algorithm in 4 is $\varepsilon$-DP and achieves excess empirical risk*

$$\mathbb{E}[\widehat{\Phi}_Z(\widehat{x}) - \widehat{\Phi}_Z^*] \leq O\left(\frac{d_x}{\varepsilon n}\left[L_{f,x}D_x + L_{f,y}D_y + \frac{L_{f,y}L_{g,y}}{\mu_g}\right]\right).$$

*Proof.* **Privacy:** First, notice that the distribution induced by the exponential weight function in 4 is the same if we use $\exp\left(-\frac{\varepsilon}{2s}[\widehat{\Phi}_Z(x) - \widehat{\Phi}_Z(x_0)]\right)$ for some arbitrary point $x_0 \in \mathcal{X}$. To establish the privacy guarantee, it suffices to show that the sensitivity of $\widehat{\Phi}_Z(x) - \widehat{\Phi}_Z(x_0)$ is upper bounded by $s$ for any $x$. Now, let $Z \sim Z'$ be any adjacent data sets differing in $z_1 \neq z_1'$ and let $x \in \mathcal{X}$. Then the sensitivity of $\widehat{\Phi}_Z(x) - \widehat{\Phi}_Z(x_0)$ is upper bounded by

$$\left|\widehat{\Phi}_Z(x) - \widehat{\Phi}_Z(x_0) - \widehat{\Phi}_{Z'}(x) + \widehat{\Phi}_{Z'}(x_0)\right|$$

$$\leq \frac{1}{n}|f(x, \widehat{y}_Z^*(x), z_1) - f(x_0, \widehat{y}_Z^*(x_0), z_1) - f(x, \widehat{y}_{Z'}^*(x), z_1') + f(x_0, \widehat{y}_{Z'}^*(x_0), z_1')|$$

$$+ \frac{1}{n}\sum_{i>1}|f(x, \widehat{y}_Z^*(x), z_i) - f(x, \widehat{y}_{Z'}^*(x), z_i)|$$

$$+ \frac{1}{n}\sum_{i>1}|f(x_0, \widehat{y}_Z^*(x_0), z_i) - f(x_0, \widehat{y}_{Z'}^*(x_0), z_i)|$$

$$\leq \frac{2}{n}[L_{f,x}D_x + L_{f,y}D_y] + \frac{1}{n}\sum_{i>1}|f(x, \widehat{y}_Z^*(x), z_i) - f(x, \widehat{y}_{Z'}^*(x), z_i)|$$

$$+ \frac{1}{n}\sum_{i>1}|f(x_0, \widehat{y}_Z^*(x_0), z_i) - f(x_0, \widehat{y}_{Z'}^*(x_0), z_i)|.$$

Now, for any $x$, we have

$$\|\widehat{y}_Z^*(x) - \widehat{y}_{Z'}^*(x)\| \leq \frac{2L_{g,y}}{\mu_g n},$$

by [44, 36]. Together with $L_{f,y}$-Lipschitz continuity of $f(x, \cdot, z)$, we can then obtain the desired sensitivity bound.

**Excess risk:** This is immediate from Lemma B.2 (stated below) in the convex case. For nonconvex $\widehat{\Phi}_Z$, the same excess risk bound holds up to logarithmic factors by [40].

$\square$

**Lemma B.2** (Utility Guarantee, [17, Corollary 1]). *Suppose $k > 0$ and $F$ is a convex function over the convex set $\mathcal{K} \subseteq \mathbb{R}^d$. If we sample $x$ according to distribution $\nu$ whose density is proportional to $\exp(-kF(x))$, then we have*

$$\mathbb{E}_\nu[F(x)] \leq \min_{x \in \mathcal{K}} F(x) + \frac{d}{k}.$$

Next, we turn to the $(\varepsilon, \delta)$-DP case.

**Approximate $(\varepsilon, \delta)$-DP.** We define the privacy curve first:

**Definition B.3** (Privacy Curve). Given two random variables $X, Y$ supported on some set $\Omega$, define the privacy curve $\delta(X\|Y) : \mathbb{R}_{\geq 0} \to [0, 1]$ as:

$$\delta(X\|Y)(\epsilon) = \sup_{S \subset \Omega} \Pr[Y \in S] - e^\epsilon \Pr[X \in S].$$

We have the following theorem from [24]:

**Theorem B.4** (Regularized Exponential Mechanism, [24]). *Given convex set $\mathcal{K} \subseteq \mathbb{R}^d$ and $\mu$-strongly convex functions $F, \tilde{F}$ over $\mathcal{K}$. Let $P, Q$ be distributions over $\mathcal{K}$ such that $P(x) \propto e^{-F(x)}$ and $Q(x) \propto e^{-\tilde{F}(x)}$. If $\tilde{F} - F$ is $G$-Lipschitz over $\mathcal{K}$, then for all $z \in [0, 1]$,*

$$\delta(P \parallel Q)(\epsilon) \leq \delta\left(\mathcal{N}(0, 1) \,\middle\|\, \mathcal{N}(\frac{G}{\sqrt{\mu}}, 1)\right)(\epsilon).$$

It suffices to bound the Lipschitz constant of $\widehat{\Phi}_Z(x) - \widehat{\Phi}_{Z'}(x)$. We have the following technical lemma:

**Lemma B.5.** *Let $\widehat{y}_Z^*(x) = argmin_{y \in \mathcal{Y}} \widehat{G}_Z(x, y)$ where $\widehat{G}_Z(x, y) = \frac{1}{n} \sum_{i=1}^n g(x, y, z_i)$. If $g(x, \cdot, z)$ is $\mu_g$-strongly convex in $y$ and $\|\nabla_y \widehat{G}_Z(x, y) - \nabla_y \widehat{G}_Z(x', y)\| \leq \beta_{g,xy} \|x - x'\|$ for all $x, y, z$, then*

$$\|\widehat{y}_Z^*(x) - \widehat{y}_Z^*(x')\| \leq \frac{\beta_{g,xy}}{\mu_g} \|x - x'\|.$$

*Proof.* Since $\widehat{y}_Z^*(x)$ is the minimizer of $\widehat{G}_Z(x, y)$, the first-order optimality condition gives:

$$\nabla_y \widehat{G}_Z(x, \widehat{y}_Z^*(x)) = 0$$

Similarly for $x'$:

$$\nabla_y \widehat{G}_Z(x', \widehat{y}_Z^*(x')) = 0$$

By $\mu_g$-strong convexity of $\widehat{G}_Z(x', \cdot)$ and the first-order optimality condition, we have:

$$\begin{aligned}
\mu_g \|\widehat{y}_Z^*(x) - \widehat{y}_Z^*(x')\|^2 &\leq \langle \nabla_y \widehat{G}_Z(x', \widehat{y}_Z^*(x)) - \nabla_y \widehat{G}_Z(x', \widehat{y}_Z^*(x')), \widehat{y}_Z^*(x) - \widehat{y}_Z^*(x') \rangle \\
&= \langle \nabla_y \widehat{G}_Z(x', \widehat{y}_Z^*(x)) - \nabla_y \widehat{G}_Z(x, \widehat{y}_Z^*(x)), \widehat{y}_Z^*(x) - \widehat{y}_Z^*(x') \rangle \\
&\leq \|\nabla_y \widehat{G}_Z(x', \widehat{y}_Z^*(x)) - \nabla_y \widehat{G}_Z(x, \widehat{y}_Z^*(x))\| \cdot \|\widehat{y}_Z^*(x) - \widehat{y}_Z^*(x')\|
\end{aligned}$$

$$\leq \beta_{g,xy} \|x' - x\| \cdot \|\widehat{y}_Z^*(x) - \widehat{y}_Z^*(x')\|.$$

Therefore:

$$\|\widehat{y}_Z^*(x) - \widehat{y}_Z^*(x')\| \leq \frac{\beta_{g,xy}}{\mu_g} \|x - x'\|.$$

$\square$

**Lemma B.6.** *Grant Assumption 2.1 and additionally assume* $\|\nabla_x f(x, y, z) - \nabla_x f(x, y', z)\| \leq$ $\beta_{f,xy}\|y - y'\|$ *and* $\|\nabla_y \widehat{G}_Z(x, y) - \nabla_y \widehat{G}_Z(x', y)\| \leq \beta_{g,xy}\|x - x'\|$ *for all* $x, x', y, y', z$. *Then, for any datasets* $Z, Z' \in \mathcal{Z}^n$ *differing in one element,* $\widehat{\Phi}_Z - \widehat{\Phi}_{Z'}$ *is* $2(\frac{L_{f,x}}{n} + \frac{L_{f,y}\beta_{g,xy}}{\mu_g n} + \frac{L_{g,y}\beta_{f,xy}}{n\mu_g})$*-Lipschitz.*

*Proof.* Suppose without loss of generality that $Z$ and $Z'$ differ only at the first element $z_1 \neq z_1'$. Then:

$$\begin{aligned}
&\widehat{\Phi}_Z(x) - \widehat{\Phi}_Z(x') - \widehat{\Phi}_{Z'}(x) + \widehat{\Phi}_{Z'}(x') \\
=&\frac{1}{n}[f(x, \widehat{y}_Z^*(x), z_1) - f(x', \widehat{y}_Z^*(x'), z_1)] + \frac{1}{n}[f(x, \widehat{y}_{Z'}^*(x), z_1') - f(x', \widehat{y}_{Z'}^*(x'), z_1')] \\
&+ \frac{1}{n}\sum_{i=2}^{n}[f(x, \widehat{y}_Z^*(x), z_i) - f(x', \widehat{y}_Z^*(x'), z_i)] - \frac{1}{n}\sum_{i=2}^{n}[f(x, \widehat{y}_{Z'}^*(x), z_i) - f(x', \widehat{y}_{Z'}^*(x'), z_i)].
\end{aligned}$$

For $i = 1$, we have

$$\begin{aligned}
&|f(x, \widehat{y}_Z^*(x), z_1) - f(x', \widehat{y}_Z^*(x'), z_1)| \\
\leq\ & |f(x, \widehat{y}_Z^*(x), z_1) - f(x', \widehat{y}_Z^*(x), z_1| + |f(x', \widehat{y}_Z^*(x), z_1) - f(x', \widehat{y}_Z^*(x'), z_1)| \\
\leq\ & L_{f,x}\|x - x'\| + \frac{L_{f,y}\beta_{g,xy}}{\mu_g}\|x - x'\|,
\end{aligned}$$

where the last inequality follows from Lemma B.5. The same argument works for $z_1'$.

For each $i \geq 2$ (where $z_i$ is the same in both datasets), recalling that $\|\widehat{y}_Z^*(x) - \widehat{y}_{Z'}^*(x)\| \leq \frac{2L_{g,y}}{\mu_g n}$, we have

$$\begin{aligned}
&[f(x, \widehat{y}_Z^*(x), z_i) - f(x, \widehat{y}_{Z'}^*(x), z_i)] - [f(x', \widehat{y}_Z^*(x'), z_i) - f(x', \widehat{y}_{Z'}^*(x'), z_i)] \\
&= \int_0^1 \nabla_x f(tx + (1-t)x', \widehat{y}_Z^*(tx + (1-t)x'), z_i)dt \cdot (x - x') \\
&\quad - \int_0^1 \nabla_x f(tx + (1-t)x', \widehat{y}_{Z'}^*(tx + (1-t)x'), z_i)dt \cdot (x - x')
\end{aligned}$$

Using the smoothness of $\nabla_x f$ with respect to $y$:

$$\begin{aligned}
&\left\| \int_0^1 [\nabla_x f(tx + (1-t)x', \widehat{y}_Z^*(tx + (1-t)x'), z_i) - \nabla_x f(tx + (1-t)x', \widehat{y}_{Z'}^*(tx + (1-t)x'), z_i)]dt \right\| \\
&\leq \int_0^1 \beta_{f,xy}\|\widehat{y}_Z^*(tx + (1-t)x') - \widehat{y}_{Z'}^*(tx + (1-t)x')\|dt \\
&\leq \int_0^1 \beta_{f,xy}\frac{2L_{g,y}}{n\mu_g}dt = \frac{2L_{g,y}\beta_{f,xy}}{n\mu_g}.
\end{aligned}$$

Therefore,

$$\begin{aligned}
&|[f(x, \widehat{y}_Z^*(x), z_i) - f(x, \widehat{y}_{Z'}^*(x), z_i)] - [f(x', \widehat{y}_Z^*(x'), z_i) - f(x', \widehat{y}_{Z'}^*(x'), z_i)]| \\
&\leq \frac{2L_{g,y}\beta_{f,xy}}{n\mu_g}\|x - x'\|
\end{aligned}$$

A similar analysis applies to the term involving $z_1$ and $z_1'$, with an additional constant accounting for the difference between functions.

Therefore,

$$\left| \widehat{\Phi}_Z(x) - \widehat{\Phi}_Z(x') - \widehat{\Phi}_{Z'}(x) + \widehat{\Phi}_{Z'}(x') \right| \leq 2 \left( \frac{L_{f,x}}{n} + \frac{L_{f,y}\beta_{g,xy}}{\mu_g n} + \frac{L_{g,y}\beta_{f,xy}}{n\mu_g} \right) \|x - x'\|,$$

as desired. $\qquad\qquad\qquad\qquad\qquad\qquad\qquad\qquad\qquad\qquad\qquad\qquad\qquad\qquad\qquad\square$

### B.2 Generalization error of the regularized exponential mechanism for bilevel SCO

Another advantage of the Regularized Exponential Mechanism is that it can have a good generalization error.

**Lemma B.7.** *If we sample the solution from density $\pi_Z(x) \propto \exp(-k(\widehat{\Phi}_Z(x) + \mu\|x\|^2/2)$, the excess population loss is bounded as*

$$\mathbb{E}_{x \sim \pi_Z, Z \sim P^n}[\Phi(x) - \Phi^*] \lesssim \frac{d_x}{k} + \mu D_x^2 + \frac{L_{f,y}L_{g,y}}{\mu_g n} + \frac{L}{\mu}\left( \frac{L_{f,y}\beta_{g,xy}}{\mu_g} + L_{f,x} \right) + \frac{L_{f,y}}{\mu_g\sqrt{k\mu}}\beta_{g,xy}(1 + \kappa_g)$$
$$+ \frac{L_{f,y}CL\sqrt{n}}{\mu_g\mu} + \frac{L_{f,y}L_{g,y}(1 + \kappa_g)}{\mu_g\sqrt{n}}$$

*where*

$$\kappa_g := \beta_{g,yy}/\mu_g \qquad\qquad\qquad\qquad\qquad\qquad\qquad (13)$$

$$L := \frac{2}{n}\left( L_{f,x} + \frac{L_{f,y}\beta_{g,xy}}{\mu_g} + \frac{L_{g,y}\beta_{g,xy}}{\mu_g} \right) \qquad\qquad (14)$$

$$C := \beta_{g,xy}(1 + \kappa_g). \qquad\qquad\qquad\qquad\qquad\qquad (15)$$

*Proof.* We have

$$\mathbb{E}_{x \sim \pi_Z, Z \sim P^n}[\Phi(x) - \Phi^*] = \mathbb{E}_{x \sim \pi_Z, Z \sim P^n}[\Phi(x) - \widehat{\Phi}_Z(x) + \widehat{\Phi}_Z(x) - \widehat{\Phi}_Z^*].$$

By Lemma B.2, we have

$$\mathbb{E}_{x \sim \pi_Z, Z \sim P^n}[\widehat{\Phi}_Z(x) - \widehat{\Phi}_Z^*] \leq \frac{d_x}{k} + \mu D_x^2.$$

Next we bound the generalization error. Note that

$$\mathbb{E}_{x \sim \pi_Z, Z \sim P^n}[\Phi(x) - \widehat{\Phi}_Z(x)] = \mathbb{E}_{x \sim \pi_Z, Z \sim P^n}[F(x, y^*(x)) - \widehat{F}_Z(x, \widehat{y}_Z^*(x))].$$

Recall that $Z = \{z_1, \cdots, z_n\}$. Suppose we replace $z_i \in Z$ by a fresh independent sample $z_i'$ from $P$ and get $Z'$. Then we have

$$\mathbb{E}_{x \sim \pi_Z, Z, z_i'}[F(x, y^*(x)) - \widehat{F}_Z(x, \widehat{y}_Z^*(x))]$$
$$= \mathbb{E}_{x \sim \pi_Z, Z, z_i'}[f(x, y^*(x), z_i') - f(x, \widehat{y}_Z^*(x), z_i)]$$
$$= \mathbb{E}_{x' \sim \pi_{Z'}, Z, z_i'}[f(x', y^*(x'), z_i)] - \mathbb{E}_{x \sim \pi_Z, Z, z_i'}[f(x, \widehat{y}_Z^*(x), z_i)]$$
$$= \mathbb{E}_{x' \sim \pi_{Z'}, Z, z_i'}[f(x', y^*(x'), z_i) - f(x', \widehat{y}_{Z'}^*(x'), z_i)]$$
$$\quad + \mathbb{E}_{x' \sim \pi_{Z'}, Z, z_i'}[f(x', \widehat{y}_{Z'}^*(x'), z_i)] - \mathbb{E}_{x' \sim \pi_{Z'}, Z, z_i'}[f(x', \widehat{y}_Z^*(x), z_i)]$$
$$\quad + \mathbb{E}_{x' \sim \pi_{Z'}, Z, z_i'}[f(x', \widehat{y}_Z^*(x'), z_i)] - \mathbb{E}_{x \sim \pi_Z, Z, z_i'}[f(x, \widehat{y}_Z^*(x), z_i)],$$

where the first equality follows by the argument in [12, Lemma 7]. Recall that in the proof of Theorem 3.2, we show that for any $x$,

$$\|\widehat{y}_Z^*(x) - \widehat{y}_{Z'}^*(x)\| \leq \frac{2L_{g,y}}{\mu_g n},$$

which leads to that

$$\mathbb{E}_{x \sim \pi_{Z'}, Z, z_i'}[f(x, \widehat{y}_{Z'}^*(x), z_i)] - \mathbb{E}_{x \sim \pi_{Z'}, Z, z_i'}[f(x, \widehat{y}_Z^*(x), z_i)] \leq \frac{2L_{f,y}L_{g,y}}{\mu_g n}.$$

By Lemma B.5, we know for any $Z, z_i$, we have

$$|f(x, \widehat{y}_Z^*(x), z_i) - f(x', \widehat{y}_Z^*(x'), z_i)| = |f(x, \widehat{y}_Z^*(x), z_i) - f(x, \widehat{y}_Z^*(x'), z_i) + f(x, \widehat{y}_Z^*(x'), z_i) - f(x', \widehat{y}_Z^*(x'), z_i)|$$

$$\leq \frac{L_{f,y}\beta_{g,xy}}{\mu_g}\|x - x'\| + L_{f,x} \cdot \|x - x'\| = \left(\frac{L_{f,y}\beta_{g,xy}}{\mu_g} + L_{f,x}\right)\|x - x'\|.$$

Moreover, by Lemma B.6 and [24], we can show that $W_2(\pi_Z, \pi_{Z'}) \leq \frac{L}{\mu}$ with $L = 2(\frac{L_{f,x}}{n} + \frac{L_{f,y}\beta_{g,xy}}{\mu_g n} + \frac{L_{g,y}\beta_{f,xy}}{n\mu_g})$. Hence we have

$$\mathbb{E}_{x\sim\pi_{Z'},Z,z_i'}[f(x, \widehat{y}_Z^*(x), z_i)] - \mathbb{E}_{x\sim\pi_Z,Z,z_i'}[f(x, \widehat{y}_Z^*(x), z_i)]$$

$$= \mathbb{E}_{Z,z_i'}[\mathbb{E}_{x\sim\pi_{Z'}}h_{Z,z_i}(x) - \mathbb{E}_{x\sim\pi_Z}h_{Z,z_i}(x)]$$

$$\leq (\frac{L_{f,y}\beta_{g,xy}}{\mu_g} + L_{f,y}) \cdot W_2(\pi_Z, \pi_{Z'})$$

$$\leq (\frac{L_{f,y}\beta_{g,xy}}{\mu_g} + L_{f,y})L/\mu.$$

Now, by Lipschitz continuity, it remains to bound the right-hand side of

$$\mathbb{E}_{x'\sim\pi_{Z'},Z,z_i'}[f(x', y^*(x), z_i) - f(x', \widehat{y}_{Z'}^*(x), z_i)] = \mathbb{E}_{x\sim\pi_Z,Z,z_i'}[f(x, y^*(x), z_i') - f(x, \widehat{y}_Z^*(x), z_i')]$$

(16)

$$\leq L_{f,y}\mathbb{E}_{x\sim\pi_Z,Z,z_i'}\|y^*(x) - \widehat{y}_Z^*(x)\|. \qquad (17)$$

For any $x$, by the strong convexity of $g$, we have

$$\langle\nabla_y G(x, \widehat{y}_Z^*(x)) - \nabla_y G(x, y^*(x)), \widehat{y}_Z^*(x) - y^*(x)\rangle \geq \mu_g\|\widehat{y}_Z^*(x) - y^*(x)\|^2.$$

By the first-order optimality, we know that $\nabla_y G(x, y^*(x)) = 0$, and hence we know

$$\|\widehat{y}_Z^*(x) - y^*(x)\| \leq \|\nabla_y G(x, \widehat{y}_Z^*(x))\|/\mu_g.$$

Similarly, we have

$$\nabla_y \widehat{G}_Z(x, \widehat{y}_Z^*(x)) = 0$$

and

$$\|\nabla_y G(x, \widehat{y}_Z^*(x))\| = \|\nabla_y G(x, \widehat{y}_Z^*(x)) - \nabla_y \widehat{G}_Z(x, \widehat{y}_Z^*(x))\|.$$

Next, we bound

$$\mathbb{E}_{x\sim\pi_Z,Z}\|\nabla_y G(x, \widehat{y}_Z^*(x)) - \nabla_y \widehat{G}_Z(x, \widehat{y}_Z^*(x))\|.$$

Let $v(Z, x) := \nabla_y G(x, \widehat{y}_Z^*(x)) - \nabla_y \widehat{G}_Z(x, \widehat{y}_Z^*(x))$. Using the law of total variance, we have

$$\mathbb{E}[\|v(Z, x)\|^2] = \underbrace{\mathbb{E}_Z[\text{Var}_x(v(Z, x))]}_{\text{Term A}} + \underbrace{\text{Var}_Z(\mathbb{E}_x[v(Z, x)])}_{\text{Term B}} + \underbrace{\|\mathbb{E}_{x,Z}[v(Z, x)]\|^2}_{\text{Term C}}.$$

Note that we have

$$\|\nabla_y G(x, \widehat{y}_Z^*(x)) - \nabla_y G(x', \widehat{y}_Z^*(x'))\|$$

$$\leq \|\nabla_y G(x, \widehat{y}_Z^*(x)) - \nabla_y G(x', \widehat{y}_Z^*(x))\| + \|\nabla_y G(x', \widehat{y}_Z^*(x)) - \nabla_y G(x', \widehat{y}_Z^*(x'))\|$$

$$\leq \beta_{g,xy}\|x - x'\| + \beta_{g,yy}\|\widehat{y}_Z^*(x) - \widehat{y}_Z^*(x')\|$$

$$\leq (\beta_{g,xy} + \frac{\beta_{g,yy}\beta_{g,xy}}{\mu_g})\|x - x'\|, \qquad (18)$$

where we apply Lemma B.5 for the last step. Similarly, we also have

$$\|\nabla_y \widehat{G}_Z(x, \widehat{y}_Z^*(x)) - \nabla_y \widehat{G}_Z(x', \widehat{y}_Z^*(x'))\| \leq (\beta_{g,xy} + \frac{\beta_{g,yy}\beta_{g,xy}}{\mu_g})\|x - x'\|.$$

For Term A, for any dataset $Z$, $\pi_Z$ is $k\mu$-strongly log-concave, and function $v$ is Lipschitz in $x$ with Lipschitz constant $(\beta_{g,xy} + \frac{\beta_{g,yy}\beta_{g,xy}}{\mu_g})$ specified in Equation 18. By Poincaré inequality, we have

$$\mathrm{Var}_{x\sim\pi_Z}(v(Z,x)) \leq \frac{1}{k\mu}\mathbb{E}[\|\nabla_x v(Z,x)\|^2] \leq \frac{1}{k\mu}(\beta_{g,xy} + \frac{\beta_{g,yy}\beta_{g,xy}}{\mu_g})^2.$$

As for the Term B, let $v(Z) = \mathbb{E}_{x\sim\pi_Z}v(Z,x)$. By the Efron-Stein Inequality, we have

$$\mathrm{Var}(v(Z)) \leq \frac{1}{2}\sum_{j\in[n]}\mathbb{E}[\|v(Z) - v(Z'_j)\|^2] = \frac{n}{2}\mathbb{E}\|v(Z) - v(Z')\|^2,$$

where $Z'_j$ is the dataset with $z_j$ replaced by a fresh sample. Then we have

$$\mathbb{E}\|v(Z) - v(Z')\|^2 = \mathbb{E}_{Z,z'_i}\|\mathbb{E}_{x\sim\pi_Z}\nabla_y G(x, \widehat{y}^*_Z(x)) - \mathbb{E}_{x\sim\pi_{Z'}}\nabla_y G(x, \widehat{y}^*_{Z'}(x))$$
$$- (\mathbb{E}_{x\sim\pi_Z}\nabla_y \widehat{G}_Z(x, \widehat{y}^*_Z(x)) - \mathbb{E}_{x\sim\pi_{Z'}}\nabla_y \widehat{G}_Z(x, \widehat{y}^*_{Z'}(x))\|^2.$$

By Kantorovich-Rubinstein duality and Cauchy-Schwartz, we have

$$\|\mathbb{E}_{x\sim\pi}h(x) - \mathbb{E}_{x'\sim\pi'}h(x')\| \leq W_1(\pi,\pi') \leq W_2(\pi,\pi')$$

for any 1-Lipschitz function $h$ and any distributions $\pi,\pi' \in L_2$. By Lemma B.5, the above fact implies

$$\|\mathbb{E}_{x\sim\pi_Z}\widehat{y}^*_Z(x) - \mathbb{E}_{x'\sim\pi_{Z'}}\widehat{y}^*_Z(x')\| \leq \frac{\beta_{g,xy}}{\mu_g}W_2(\pi_Z,\pi_{Z'}) \leq \frac{\beta_{g,xy}}{\mu_g}L/\mu.$$

A similar argument can be used to bound

$$\|\mathbb{E}_{x\sim\pi_Z}\nabla_y G(x,\widehat{y}^*_Z(x)) - \mathbb{E}_{x'\sim\pi_{Z'}}\nabla_y G(x',\widehat{y}^*_Z(x'))\| \leq \beta_{g,xy}W_2(\pi_Z,\pi_{Z'}) \leq \beta_{g,xy}L/\mu,$$

and likewise for $\nabla_y\widehat{G}_Z(x,\widehat{y}^*_Z(x))$. Hence

$$\mathbb{E}\|v(Z) - v(Z')\|^2/100 \leq \mathbb{E}_{Z,z'_i}\|\mathbb{E}_{x\sim\pi_Z}\nabla_y G(x,\widehat{y}^*_Z(x)) - \mathbb{E}_{x'\sim\pi_{Z'}}\nabla_y G(x',\widehat{y}^*_Z(x'))\|^2$$
$$+ \mathbb{E}_{Z,z'_i}\|\mathbb{E}_{x'\sim\pi_{Z'}}\nabla_y G(x',\widehat{y}^*_Z(x')) - \mathbb{E}_{x'\sim\pi_{Z'}}\nabla_y G(x',\widehat{y}^*_{Z'}(x'))\|^2$$
$$+ \mathbb{E}_{Z,z'_i}\|\mathbb{E}_{x'\sim\pi_{Z'}}\nabla_y \widehat{G}_{Z'}(x',\widehat{y}^*_{Z'}(x')) - \mathbb{E}_{x\sim\pi_Z}\nabla_y\widehat{G}_{Z'}(x,\widehat{y}^*_{Z'}(x))\|^2$$
$$+ \mathbb{E}_{Z,z'_i}\|\mathbb{E}_{x\sim\pi_Z}\nabla_y\widehat{G}_Z(x,\widehat{y}^*_Z(x)) - \mathbb{E}_{x\sim\pi_Z}\nabla_y\widehat{G}_{Z'}(x,\widehat{y}^*_{Z'}(x))\|^2$$
$$\leq \left(\beta_{g,xy}\frac{L}{\mu}\right)^2 + \left(\beta_{g,yy}\left(\frac{\beta_{g,xy}}{\mu_g}\frac{L}{\mu} + \frac{L_{g,y}}{\mu_g n}\right)\right)^2$$
$$+ \left(\left(\beta_{g,xy} + \frac{\beta_{g,xy}\beta_{g,yy}}{\mu_g}\right)\frac{L}{\mu}\right)^2 + \left(\frac{L_{g,y}\beta_{g,yy}}{\mu_g n}\right)^2 + \frac{L^2_{g,y}}{n^2}.$$

Therefore,

$$\mathrm{Var}(v(Z)) \lesssim C^2 n\left(\frac{L}{\mu}\right)^2 + \frac{L^2_{g,y}(\kappa_g + 1)^2}{n} \tag{19}$$

for $C$ defined in the lemma statement.

We already bounded $\mathrm{Var}(v(Z))$. As $\mathbb{E}\|v(Z)\|^2 = \mathrm{Var}(v(Z)) + \|\mathbb{E}v(Z)\|^2$. It remains to bound Term C: $\|\mathbb{E}v(Z)\|^2$. For this, we have

$$\mathbb{E}v(Z) = \mathbb{E}_{x\sim\pi_Z,Z,z'_i}[\nabla_y g(x,\widehat{y}^*_Z(x),z'_i) - \nabla_y g(x,\widehat{y}^*_Z(x),z_i)]$$
$$= \mathbb{E}_{x'\sim\pi_{Z'},Z,z'_i}[\nabla_y g(x',\widehat{y}^*_{Z'}(x'),z_i) - \mathbb{E}_{x\sim\pi_Z,Z,z'_i}[\nabla_y g(x,\widehat{y}^*_Z(x),z_i)]$$
$$= \mathbb{E}_{x'\sim\pi_{Z'},Z,z'_i}[\nabla_y g(x',\widehat{y}^*_{Z'}(x'),z_i) - \mathbb{E}_{x'\sim\pi_{Z'},Z,z'_i}[\nabla_y g(x',\widehat{y}^*_Z(x'),z_i]$$
$$+ \mathbb{E}_{x'\sim\pi_{Z'},Z,z'_i}[\nabla_y g(x',\widehat{y}^*_Z(x'),z_i) - \mathbb{E}_{x\sim\pi_Z,Z,z'_i}[\nabla_y g(x,\widehat{y}^*_Z(x),z_i)].$$

Hence

$$\|\mathbb{E}v(Z)\| \leq \beta_{g,yy}\frac{2L_{g,y}}{\mu_g n} + (\beta_{g,xy} + \frac{\beta_{g,yy}\beta_{g,xy}}{\mu_g})W_2(\pi_Z,\pi_{Z'})$$
$$\leq \beta_{g,yy}\frac{2L_{g,y}}{\mu_g n} + (\beta_{g,xy} + \frac{\beta_{g,yy}\beta_{g,xy}}{\mu_g})\frac{L}{\mu}.$$

Taking square roots and combining all the pieces above yields the lemma. $\qquad\square$

**Theorem B.8** (Precise Statement of Theorem 3.3). *Grant Assumption 2.1 and parts 3 and 4 of Assumption 2.2. Assume $\widehat{\Phi}_Z$ and $\Phi$ are convex for all $Z$. Sampling $\widehat{x}$ from a distribution proportional to $\exp(-k(\widehat{\Phi}_Z(x) + \mu\|x\|^2/2))$ with $k = O\left(\frac{\mu n^2 \varepsilon^2}{G^2 \log(1/\delta)}\right)$ and $G = (L_{f,x} + \frac{L_{f,y}\beta_{g,xy}}{\mu_g} + \frac{L_{g,y}\beta_{f,xy}}{\mu_g})$ is $(\varepsilon, \delta)$-DP. Moreover,*

- *setting $\mu = \frac{G\sqrt{d_x \log(1/\delta)}}{nD_x\varepsilon}$, we achieve excess risk*

$$\mathbb{E}\widehat{\Phi}_Z(\widehat{x}) - \widehat{\Phi}_Z^* \leq O\left(\left(L_{f,x} + \frac{L_{f,y}\beta_{g,xy}}{\mu_g} + \frac{L_{g,y}\beta_{f,xy}}{\mu_g}\right)D_x\frac{\sqrt{d_x\log(1/\delta)}}{n\varepsilon}\right).$$

- *setting*

$$\mu = \frac{1}{D_x}\sqrt{\frac{G}{\varepsilon n}\frac{L_{f,y}C}{\mu_g} + G^2\frac{d_x\log(1/\delta)}{\varepsilon^2 n^2} + \frac{GCL_{f,y}}{\mu_g\sqrt{n}} + \frac{G}{n}\left(L_{f,x} + \frac{L_{f,y}\beta_{g,xy}}{\mu_g}\right)},$$

*for $C := \beta_{g,xy}(1 + \kappa_g)$ with $\kappa_g = \beta_{g,yy}/\mu_g$, the population loss has the following guarantee:*

$$\mathbb{E}\Phi(\widehat{x}) - \Phi^* \lesssim D_x\left[G\frac{\sqrt{d_x\log(1/\delta)}}{\varepsilon n} + \frac{1}{\sqrt{n}}\left(\sqrt{\frac{GL_{f,y}C}{\mu_g\varepsilon}} + \sqrt{G(L_{f,x} + L_{f,y}\beta_{g,xy}/\mu_g)} + L_{g,y}(\kappa_g + 1)\frac{L_{f,y}}{\mu_g}\right)\right.$$

$$\left. + \frac{1}{n^{1/4}}\sqrt{\frac{GCL_{f,y}}{\mu_g}} + \frac{L_{f,y}L_{g,y}\kappa_g}{\mu_g n}\right]$$

*Proof.* The privacy guarantee follows from the privacy curve of Gaussian variables and Lemma 6.3 in [24].

When setting $\mu = \frac{G\sqrt{d\log(1/\delta)}}{nD_x\varepsilon}$, Lemma B.2 gives us that

$$\mathbb{E}\widehat{\Phi}_Z(\widehat{x}) - \widehat{\Phi}_Z^* \lesssim \frac{d_x}{k} + \frac{\mu D_x^2}{2} = \frac{d_x G^2\log(1/\delta)}{\mu n^2\varepsilon^2} + \frac{\mu D_x^2}{2} = O\left(GD_x\frac{\sqrt{d_x\log(1/\delta)}}{n\varepsilon}\right).$$

The population loss guarantee follows from plugging the prescribed $\mu$ and $k$ into Lemma B.7. $\qquad\square$

## B.3 Efficient implementation of conceptual algorithms

In many practical applications of optimization and sampling algorithms, we face unavoidable approximation errors when evaluating functions. Given any $x$, we may not get the exact $\widehat{y}_Z^*(x)$ in solving the low-level optimization, which means we may introduce a small error each time we compute the function value of $f(x, \widehat{y}_Z^*(x), z)$. This section analyzes how such small function evaluation errors affect log-concave sampling algorithms. We establish bounds on the impact of errors bounded by $\zeta$ on the conductance, mixing time, and distributional accuracy of Markov chains used for sampling. We then develop an efficient implementation based on the [9] approach that maintains polynomial time complexity while providing formal guarantees on sampling accuracy in the presence of function evaluation errors.

### B.3.1 Original Grid-Walk Algorithm for Log-Concave Sampling

We first state the classic Grid-Walk algorithm from Applegate and Kannan [3] on sampling from log-concave distributions.

Let $F(\cdot)$ be a real positive-valued function defined on a cube $A = [a, b]^d$ in $\mathbb{R}^d$, where $[a, b]^d$ represents a hypercube with side length $\kappa := b - a$. Let $f(\theta) = -\log F(\theta)$ and suppose there exist real numbers $\alpha, \beta$ such that:

$$|f(x) - f(y)| \leq \alpha \left( \max_{i \in [1,d]} |x_i - y_i| \right),$$

$$f(\lambda x + (1 - \lambda)y) \geq \lambda f(x) + (1 - \lambda)f(y) - \beta,$$

for all $x, y \in A$ and $\lambda \in [0, 1]$.

Let $\gamma \leq 1/(2\alpha)$ be a discretization parameter. The following algorithm samples from a distribution $\nu$ on the continuous domain $A$ such that for all $\theta \in A$, $|\nu(\theta) - cF(\theta)| \leq \epsilon$, where $c$ is a normalization constant:

1. Divide the cube $A$ into small cubes $\{C_x\}$ of side length $\gamma$, with centers $\{x\}$. Let $\Omega$ be the set of all such centers.

2. If $\kappa < 1/\alpha$, then pick a point $\theta$ uniformly from $A$ and output $\theta$ with probability $F(\theta)/(e \max_{x \in A} F(x))$; otherwise restart.

3. For $\kappa \geq 1/\alpha$, proceed as follows:

    (a) Choose a starting point $x_0 \in \Omega$ arbitrarily.
    (b) Define a random walk on the centers of the small cubes as follows:
        i. At a state (cube center) $x$, stay at $x$ with probability 1/2.
        ii. Otherwise (with probability 1/2), choose a direction $u \in \{\pm e_1, \cdots, \pm e_d\}$ uniformly at random (each chosen with probability $1/2d$), where $e_i$ is the standard basis vector in the $i$-th coordinate.
        iii. If the adjacent cube in that direction is not in $A$, stay at $x$.
        iv. Otherwise, move to the center $y$ of that adjacent cube with probability $\min\{1, F(y)/F(x)\}$; with probability $1 - \min\{1, F(y)/F(x)\}$, remain at $x$.
    (c) Run this random walk for $T$ steps. Let $x$ be the final state.
    (d) Pick a point $\theta$ uniformly from the cube $C_x$.
    (e) Output $\theta$ with probability $F(\theta)/(eF(x))$; otherwise, restart from step 3(a) with a new recursive call.

For implementation details, we refer to the original paper [3]. In the subsections that follow, we analyze how this algorithm behaves when the function $F$ can only be evaluated with some bounded error, a common scenario in practical applications.

### B.3.2 Conductance Bound with Function Evaluation Errors

The conductance of a Markov chain measures how well the chain mixes, specifically how quickly it converges to its stationary distribution. For a Markov chain with state space $\Omega$, transition matrix $P$

and stationary distribution $q$, the conductance $\varphi$ is defined as:

$$\varphi := \min_{S \subset \Omega : 0 < q(S) \le 1/2} \frac{\sum_{x \in S, y \in \Omega \setminus S} q(x) P_{xy}}{q(S)}.$$

Higher conductance implies faster mixing, while lower conductance suggests the presence of bottlenecks in the state space.

We now analyze how small errors in function evaluation affect the conductance of the Markov chain used in log-concave sampling, described in Section B.3.1. This analysis is central to understanding the robustness of sampling algorithms in the presence of approximation errors.

**Lemma B.9** (Re-statement of Lemma 3.6). *Let $P$ be the transition matrix of the original Markov chain in the grid-walk algorithm of Section B.3.1 based on function $f$, with state space $\Omega$ and conductance $\varphi$. Let $P'$ be the transition matrix of the perturbed chain based on $f'$ where $f'(\theta) = f(\theta) + \zeta(\theta)$ with $|\zeta(\theta)| \le \zeta$ for all $\theta \in \Omega$, where $\zeta(\cdot)$ is an arbitrary bounded error function and $\zeta > 0$ is an upper bound on its magnitude. Then the conductance $\varphi'$ of the perturbed chain satisfies:*

$$\varphi' \ge e^{-6\zeta} \varphi.$$

*Proof.* Fix any subset $S$ of the state space. The conductance of $S$ in the original chain is:

$$\varphi_S = \frac{\sum_{x \in S, y \notin S} q(x) P_{xy}}{\min\{\sum_{x \in S} q(x), \sum_{x \notin S} q(x)\}}.$$

where $q$ is the stationary distribution and $P_{xy}$ are the transition probabilities.

In the grid-walk algorithm, we know $P_{xy} = 0$ if $x \ne y$ are not adjacent; for adjacent points $x$ and $y$:

$$P_{xy} = \frac{1}{4d} \min\left\{1, \frac{F(y)}{F(x)}\right\} = \frac{1}{4d} \min\{1, e^{-(f(y)-f(x))}\},$$

and remarkably $P_{xx} = 1 - \sum_{x \ne y} P_{xy}$.

For the perturbed chain with adjacent $x, y$:

$$P'_{xy} = \frac{1}{4d} \min\left\{1, \frac{F'(y)}{F'(x)}\right\} = \frac{1}{4d} \min\{1, e^{-(f'(y)-f'(x))}\}.$$

Since $f'(y) - f'(x) = f(y) - f(x) + (\zeta(y) - \zeta(x))$ and $|\zeta(y) - \zeta(x)| \le 2\zeta$, we have:

$$e^{-(f(y)-f(x)-2\zeta)} \le e^{-(f'(y)-f'(x))} \le e^{-(f(y)-f(x)+2\zeta)}.$$

This implies:

$$e^{-2\zeta} \min\{1, e^{-(f(y)-f(x))}\} \le \min\{1, e^{-(f'(y)-f'(x))}\} \le e^{2\zeta} \min\{1, e^{-(f(y)-f(x))}\}.$$

Therefore:

$$e^{-2\zeta} P_{xy} \le P'_{xy} \le e^{2\zeta} P_{xy}.$$

The stationary distributions $q$ and $q'$ satisfy:

$$q(x) = \frac{F(x)}{\sum_{z \in \Omega} F(z)} \quad \text{and} \quad q'(x) = \frac{F'(x)}{\sum_{z \in \Omega} F'(z)}.$$

Since $F'(x) = e^{-f'(x)} = e^{-(f(x)+\zeta(x))} = e^{-f(x)} e^{-\zeta(x)} = F(x) e^{-\zeta(x)}$, we have:

$$e^{-\zeta} \cdot q(x) \cdot \frac{\sum_{z \in \Omega} F(z)}{\sum_{z \in \Omega} F'(z)} \le q'(x) \le e^{\zeta} \cdot q(x) \cdot \frac{\sum_{z \in \Omega} F(z)}{\sum_{z \in \Omega} F'(z)}.$$

The normalization ratio satisfies:

$$e^{-\zeta} \le \frac{\sum_{z \in \Omega} F'(z)}{\sum_{z \in \Omega} F(z)} \le e^{\zeta}.$$

Therefore:

$$e^{-2\zeta} \cdot q(x) \leq q'(x) \leq e^{2\zeta} \cdot q(x).$$

Using the bounds on transition probabilities and stationary distributions:

$$\sum_{x \in S, y \notin S} q'(x) P'_{xy} \geq e^{-4\zeta} \sum_{x \in S, y \notin S} q(x) P_{xy}.$$

And:

$$\min \left\{ \sum_{x \in S} q'(x), \sum_{x \notin S} q'(x) \right\} \leq e^{2\zeta} \min \left\{ \sum_{x \in S} q(x), \sum_{x \notin S} q(x) \right\}.$$

Therefore:

$$\begin{aligned}
\varphi'_S &= \frac{\sum_{x \in S, y \notin S} q'(x) P'_{xy}}{\min\{\sum_{x \in S} q'(x), \sum_{x \notin S} q'(x)\}} \\
&\geq \frac{e^{-4\zeta} \sum_{x \in S, y \notin S} q(x) P_{xy}}{e^{2\zeta} \min\{\sum_{x \in S} q(x), \sum_{x \notin S} q(x)\}} \\
&= e^{-6\zeta} \varphi_S.
\end{aligned}$$

Since $\varphi = \min_S \varphi_S$ and $\varphi' = \min_S \varphi'_S$, we have:

$$\varphi' \geq e^{-6\zeta} \varphi.$$

$\square$

### B.3.3 Relative Distance Bound Between $F$ and $F'$

We now analyze how function evaluation errors affect the distributional distance between the original and perturbed stationary distributions.

For distributions, we define the $L_\infty$ distance (or log-ratio distance) between distributions $\mu$ and $\nu$ on $A$ as:

$$\text{Dist}_\infty(\mu, \nu) = \sup_{\theta \in A} \left| \log \frac{\mu(\theta)}{\nu(\theta)} \right|. \tag{20}$$

**Lemma B.10** (Re-statement of Lemma 3.7). *Let $F(\theta) = e^{-f(\theta)}$ and $F'(\theta) = e^{-f'(\theta)}$ where $f'(\theta) = f(\theta) + \zeta(\theta)$ with $|\zeta(\theta)| \leq \zeta$ for all $\theta \in A$. Then the relative distance between $F$ and $F'$ is bounded by:*

$$e^{-\zeta} \leq \frac{F'(\theta)}{F(\theta)} \leq e^\zeta, \quad \forall \theta \in A.$$

*Furthermore, if we define the distributions $\pi(\theta) \propto F(\theta)$ and $\pi'(\theta) \propto F'(\theta)$, then the infinity-distance between them is bounded by:*

$$\text{Dist}_\infty(\pi', \pi) = \sup_{\theta \in A} \left| \log \frac{\pi'(\theta)}{\pi(\theta)} \right| \leq 2\zeta.$$

*Proof.* For any $\theta \in A$, we have:

$$F'(\theta) = e^{-f'(\theta)} = e^{-(f(\theta) + \zeta(\theta))} = e^{-f(\theta)} e^{-\zeta(\theta)} = F(\theta) e^{-\zeta(\theta)}.$$

Since $|\zeta(\theta)| \leq \zeta$, we have:

$$e^{-\zeta} \leq e^{-\zeta(\theta)} \leq e^\zeta.$$

Therefore:

$$e^{-\zeta} F(\theta) \leq F'(\theta) \leq e^{\zeta} F(\theta).$$

For the normalized distributions, we have:

$$\pi(\theta) = \frac{F(\theta)}{\int_A F(z)dz},$$

$$\pi'(\theta) = \frac{F'(\theta)}{\int_A F'(z)dz} = \frac{F(\theta)e^{-\zeta(\theta)}}{\int_A F(z)e^{-\zeta(z)}dz}.$$

This gives:

$$\frac{\pi'(\theta)}{\pi(\theta)} = \frac{F(\theta)e^{-\zeta(\theta)}}{\int_A F(z)e^{-\zeta(z)}dz} \cdot \frac{\int_A F(z)dz}{F(\theta)} = e^{-\zeta(\theta)} \cdot \frac{\int_A F(z)dz}{\int_A F(z)e^{-\zeta(z)}dz}.$$

Since $e^{-\zeta} \leq e^{-\zeta(z)} \leq e^{\zeta}$ for all $z \in A$, we have:

$$e^{-\zeta} \int_A F(z)dz \leq \int_A F(z)e^{-\zeta(z)}dz \leq e^{\zeta} \int_A F(z)dz.$$

Therefore:

$$e^{-2\zeta} \leq \frac{\pi'(\theta)}{\pi(\theta)} \leq e^{2\zeta}.$$

Thus, the $L_\infty$-distance between $\pi'$ and $\pi$ is bounded by:

$$\mathrm{Dist}_\infty(\pi', \pi) = \sup_{\theta \in A} \left| \ln \frac{\pi'(\theta)}{\pi(\theta)} \right| \leq 2\zeta.$$

$\square$

### B.3.4 Mixing Time Analysis and Implementation Details

For a Markov chain with state space $\Omega$, transition matrix $P$, and stationary distribution $\pi$, the mixing time $t_{\mathrm{mix}}(\epsilon)$ with respect to the $L_\infty$-distance is defined as:

$$t_{\mathrm{mix}}(\epsilon) = \min\{t \geq 0 : \max_{x \in \Omega} \mathrm{Dist}_\infty(P^t(x, \cdot), \pi(\cdot)) \leq \epsilon\}, \tag{21}$$

for any $\epsilon \geq 0$. For efficient implementation of the grid-walk algorithm, we utilize the results of [9]. Following their approach, we can determine the number of steps required for $L_\infty$ convergence using:

**Lemma B.11** (Mixing time for relative $L_\infty$ convergence [41]). *Let $P$ be a lazy, time-reversible Markov chain over a finite state space $\Gamma$ with stationary distribution $\pi$. Then, the mixing time of $P$ w.r.t. $L_\infty$ distance is at most*

$$t_\infty \leq 1 + \int_{4\pi^*}^{4/\epsilon} \frac{4\mathrm{d}x}{x\varphi^2(x)} \tag{22}$$

*where $\varphi(x) = \inf\{\varphi_S : \pi(S) \leq x\}$, $\varphi_S$ denotes the conductance of the set $S \subseteq \Gamma$, and $\pi_* = \min_{x \in \Gamma} \pi(x)$ is the minimum probability assigned by the stationary distribution.*

We now provide a bound on how function evaluation errors affect the mixing time.

**Lemma B.12** (Re-statement of Lemma 3.8). *The mixing time $t'_{mix}(\epsilon)$ of the perturbed chain to achieve $L_\infty$-distance $\epsilon$ to its stationary distribution satisfies:*

$$t'_{mix}(\epsilon) \leq e^{12\zeta} \cdot O\left( \frac{\alpha^2 \tau^2 d^2}{\epsilon^2} e^{\epsilon} \max\left\{ d \log \frac{\alpha\tau\sqrt{d}}{\epsilon}, \alpha\tau \right\} \right).$$

*Proof.* For a log-concave function $F(x) = e^{-f(x)}$ where $f$ is $\alpha$-Lipschitz, we set the grid spacing parameter $\gamma = \frac{\epsilon}{2\alpha\sqrt{d}}$. Using the conductance bound from previous analysis [9], we can derive the lower bound on conductance:

$$\varphi(x) \geq \frac{\epsilon}{8\alpha\tau d^{3/2}e^\epsilon}.$$

By Lemma 3.6, the conductance $\varphi'$ for the perturbed chain $F'$ satisfies

$$\varphi'(x) \geq \frac{e^{-6\zeta}\epsilon}{8\alpha\tau d^{3/2}e^\epsilon}.$$

By the Lipschitz assumption, we have that

$$\pi'(u) = \frac{F'(u)}{\sum_{v\in\Omega} F'(v)} \geq \frac{e^{-\alpha\tau-2\zeta}}{(\tau/\gamma)^d}.$$

Hence, by the lower bounds of conductance and minimum probability in the state space and Lemma B.11, we complete the proof.

$\square$

Building upon our analysis of how function evaluation errors affect conductance, mixing time, and distributional distance, we now develop an efficient algorithm for sampling from log-concave distributions in the presence of such errors. Our approach builds upon the framework developed by Bassily, Smith, and Thakurta [9], extending it to handle approximation errors with formal guarantees.

**Theorem B.13** (BST14-Based Implementation). *Let $C \subset \mathbb{R}^d$ be a convex set and $f : C \to \mathbb{R}$ be a convex, L-Lipschitz function. There exists an efficient algorithm that, when given exact function evaluations, outputs a sample $\theta \in C$ from a distribution $\mu$ such that the relative distance between $\mu$ and the target log-concave distribution $\pi(\theta) \propto e^{-f(\theta)}$ can be made arbitrarily small, i.e., $\mathrm{Dist}_\infty(\mu, \pi) \leq \xi$ for any desired $\xi > 0$. This algorithm runs in time $O(d^3 \cdot poly(L, \|C\|_2, 1/\xi))$, which is polynomial in the dimension $d$, the diameter of $C$, the Lipschitz constant $L$, and the accuracy parameter $1/\xi$.*

The key techniques in this implementation include:

1. Extending the function $f$ beyond the convex set $C$ to a surrounding cube $A$

2. Using a gauge penalty function to reduce the probability of sampling outside $C$

3. Implementing an efficient grid-walk algorithm to sample from the resulting distribution

We now formally incorporate the effect of function evaluation errors into this framework:

**Theorem B.14** (Re-statement of Theorem 3.9). *Let $C \subset \mathbb{R}^d$ be a convex set and $f : C \to \mathbb{R}$ be a convex, L-Lipschitz function. Suppose we have access to an approximate function evaluator that returns $f'(\theta) = f(\theta) + \zeta(\theta)$ where $|\zeta(\theta)| \leq \zeta$ for all $\theta \in C$, and $\zeta = O(1)$ is a constant independent of dimension. There exists an efficient algorithm that outputs a sample $\theta \in C$ from a distribution $\mu'$ such that:*

$$\mathrm{Dist}_\infty(\mu', \pi) \leq 2\zeta + \xi \tag{23}$$

*where $\pi(\theta) \propto e^{-f(\theta)}$ is the target log-concave distribution and $\delta > 0$ is an arbitrarily small constant.*

*This algorithm runs in time $O(e^{12\zeta} \cdot d^3 \cdot poly(L, \|C\|_2, 1/\xi))$. When $\zeta = O(1)$ is a constant, this remains $O(d^3 \cdot poly(L, \|C\|_2, 1/\xi))$ with the same asymptotic complexity as the exact evaluation algorithm, differing only by a constant factor $e^{12\zeta}$ in the running time.*

*Proof.* We follow the approach of [9] with appropriate modifications to account for function evaluation errors:

1. Enclose the convex set $C$ in an isotropic cube $A$ with edge length $\tau = \|C\|_\infty$.

2. Construct a convex Lipschitz extension $\overline{f}$ of the function $f$ over $A$ using:

$$\overline{f}(x) = \min_{y \in C} \left( f(y) + L\|x - y\|_2 \right)$$

This extension preserves the Lipschitz constant $L$ and the convexity of $f$.

3. Define a gauge penalty function using the Minkowski functional of $C$:

$$\overline{\psi}_\alpha(\theta) = \alpha \cdot \max\{0, \psi(\theta) - 1\}$$

where $\psi(\theta) := \inf\{r > 0 : \theta \in rC\}$ is the Minkowski norm of $\theta$ with respect to $C$, and $\alpha$ is a parameter set to ensure correct sampling properties.

4. Define the target sampling distribution:

$$\pi(\theta) \propto e^{-\overline{f}(\theta) - \overline{\psi}_\alpha(\theta)}, \ \forall \theta \in A.$$

5. In the presence of function evaluation errors, for $\theta \in A$, the algorithm samples from:

$$\pi'(\theta) \propto e^{-\overline{f}'(\theta) - \overline{\psi}_\alpha(\theta)}$$

where $\overline{f}'(\theta) = \overline{f}(\theta) + \zeta(\theta)$ and $|\zeta(\theta)| \leq \zeta$.

6. By Lemma 3.7 on the relative distance between distributions, we have:

$$\mathrm{Dist}_\infty(\pi', \pi) \leq 2\zeta.$$

7. For the sampling algorithm's computational efficiency, we note that by Corollary 3.8, the mixing time increases by a factor of $e^{12\zeta}$. and the modified algorithm's running time becomes $O(e^{12\zeta} \cdot d^3 \cdot \mathrm{poly}(L, \|C\|_2, 1/\xi))$.

If $\zeta$ is a constant, then the factor $e^{12\zeta}$ is also a constant. Therefore, the algorithm maintains the same asymptotic polynomial complexity in $d$ as the exact evaluation algorithm, with only the leading constant factor affected by the approximation error. $\qquad\square$

**Theorem B.15** (Exponential Mechanism Implementation). *Under Assumptions 2.1, for any constants $\varepsilon = O(1)$, there is an efficient sampler to solve DP-bilevel ERM with the following guarantees:*

- *The scheme is $(\varepsilon, 0)$-DP;*

- *The expected loss is bounded by $\tilde{O}\left( \frac{d_x}{\varepsilon n} \left[ L_{f,x} D_x + L_{f,y} D_y + \frac{L_{f,y} L_{g,y}}{\mu_g} \right] \right)$;*

- *The running time is $O\left( d^6 n \cdot \mathrm{poly}(\overline{L}, D_x, 1/\varepsilon, \log(d L_{f,y}^2/\mu_g)) \wedge d^4 n \cdot \frac{L_{g,y}}{\mu_g} \cdot \mathrm{poly}(\overline{L}, D_x, 1/\varepsilon) \right).$*

*Proof.* **Privacy:** Let $Z$ and $Z'$ be adjacent data sets. Consider the exponential mechanism and the probability density $\pi_Z$ proportional to $\exp(-\frac{\varepsilon'}{2s} \widehat{\Phi}_Z(x))$. We set $\zeta = \xi = \varepsilon'/6$. Let the $\pi'_Z$ be the probability density of the final output of the sampler. Then by Theorem B.14, we know

$$\mathrm{Dist}_\infty(\pi'_Z, \pi_Z) \leq \varepsilon'/2.$$

By Theorem 3.2, we have

$$\mathrm{Dist}_\infty(\pi_Z, \pi_{Z'}) \leq \varepsilon'.$$

Hence we know

$$\mathrm{Dist}_\infty(\pi'_Z, \pi'_{Z'}) \leq 2\varepsilon',$$

and setting $\varepsilon' = \varepsilon/2$ completes the proof of the privacy guarantee.

**Excess risk:** The excess risk bound follows from Theorem 3.2 and the assumption that $\varepsilon = O(1)$.

**Time complexity:** Given a function value query of $\widehat{\Phi}_Z(x)$, we need to return a value of error at most $\zeta$. By the Lipschitz of $f(x, \cdot, z)$, it suffices to find a point $y$ such that

$$\|y - \widehat{y}_Z^*(x)\| \leq \zeta / L_{f,y}.$$

By the strong convexity of $g(x, \cdot, z)$, there are multiple ways to find the qualified $y$. In our case, we can simply apply the cutting plane method [32], which can be implemented in $O(d^3 n \mathrm{poly}(\log(dL_{f,y}^2/\zeta\mu_g)))$. Alternatively, we could apply the subgradient method to $\widehat{G}_Z(x, \cdot)$, which can be implemented in $O(dn(\frac{L_{g,y}}{\mu_g\zeta}))$. Combining the query complexity in Theorem B.14 gives the total running time complexity. $\square$

With Theorem B.8 and a similar argument on the implementation, we can get the following result of the Regularized Exponential Mechanism.

**Theorem B.16** (Regularized Exponential Mechanism Implementation). *Grant Assumptions 2.1 and additionally assume $\|\nabla_x f(x, y, z) - \nabla_x f(x, y', z)\| \leq \beta_{f,xy}\|y - y'\|$ and $\|\nabla_y \widehat{G}_Z(x, y) - \nabla_y \widehat{G}_Z(x', y)\| \leq \beta_{g,xy}\|x - x'\|$ for all $x, x', y, y', z$. Given $\varepsilon = O(1)$ and $0 < \delta < 1/10$, there is an efficient sampler to implement the Regularized Exponential Mechanism and solve DP-bilevel ERM with the following guarantees:*

- *The scheme is $(\varepsilon, \delta)$-DP;*

- *The expected empirical loss is bounded by $O\left(\left(L_{f,x} + \frac{L_{f,y}\beta_{g,xy}}{\mu_g} + \frac{L_{g,y}\beta_{f,xy}}{\mu_g}\right) D_x \frac{\sqrt{d_x \log(1/\delta)}}{n}\right)$.*

- *The running time is $O\left(d^6 n \cdot \mathrm{poly}(\overline{L}, D_x, 1/\varepsilon, \log(dL_{f,y}^2/\mu_g)) \wedge d^4 n \cdot \frac{L_{g,y}}{\mu_g} \cdot \mathrm{poly}(\overline{L}, D_x, 1/\varepsilon)\right)$.*

*With a different parameter setting, we can get the $(\varepsilon, \delta)$-DP sampler with the same running time and achieve the expected population loss as $O\left(\left(L_{f,x} + \frac{L_{f,y}\beta_{g,xy}}{\mu_g} + \frac{L_{g,y}\beta_{f,xy}}{\mu_g}\right) D_x \left(\frac{\sqrt{d_x \log(1/\delta)}}{n} + \frac{1}{\sqrt{n}}\right)\right)$.*

## B.4 Excess risk lower bounds

**Theorem B.17** (Re-statement of Theorem 3.10). *1. Let $\mathcal{A}$ be $\varepsilon$-DP. Then, there exists a data set $Z \in \mathcal{Z}^n$ and a convex bilevel ERM problem instance satisfying Assumptions 2.1 and 2.2 with $\mu_g = \Theta(L_{g,y}/D_y)$ such that*

$$\mathbb{E}\widehat{\Phi}_Z(\mathcal{A}(Z)) - \widehat{\Phi}_Z^* = \Omega\left((L_{f,x}D_x + L_{f,y}D_y)\min\left\{1, \frac{d_x}{n\varepsilon}\right\}\right).$$

*2. Let $\mathcal{A}$ be $(\varepsilon, \delta)$-DP with $2^{-\Omega(n)} \leq \delta \leq 1/n^{1+\Omega(1)}$. Then, there exists a data set $Z \in \mathcal{Z}^n$ and a convex bilevel ERM problem instance satisfying Assumptions 2.1 and 2.2 with $\mu_g = \Theta(L_{g,y}/D_y)$ such that*

$$\mathbb{E}\widehat{\Phi}_Z(\mathcal{A}(Z)) - \widehat{\Phi}_Z^* = \Omega\left((L_{f,x}D_x + L_{f,y}D_y)\min\left\{1, \frac{\sqrt{d_x \log(1/\delta)}}{n\varepsilon}\right\}\right).$$

*Proof.* **Case 1:** Suppose $L_{f,x}D_x \lesssim L_{f,y}D_y$. Then we will show $\widehat{\Phi}_Z(\mathcal{A}(Z)) - \widehat{\Phi}_Z^* = \Omega\left((L_{f,y}D_y)\min\left\{1, \frac{d}{n\varepsilon}\right\}\right)$ with probability at least $1/2$ for pure $\varepsilon$-DP $\mathcal{A}$ and $\widehat{\Phi}_Z(\mathcal{A}(Z)) - \widehat{\Phi}_Z^* = \Omega\left((L_{f,y}D_y)\min\left\{1, \frac{\sqrt{d}}{n\varepsilon}\right\}\right)$ with probability at least $1/3$ for $(\varepsilon, \delta)$-DP $\mathcal{A}$.

Let $f(x, y, z) = -\langle y, z\rangle$, which is convex and 1-Lipschitz in $x$ and $y$ if $\mathcal{X} = \mathcal{Y} = \mathbb{B}$ are unit balls in $\mathbb{R}^d$, $d = d_x = d_y$, and $\mathcal{Z} = \{\pm 1/\sqrt{d}\}^d$. Let $g(x, y, z) = \frac{1}{2}\|y - \zeta x\|^2$ for $\zeta > 0$ to be chosen later. Note $\widehat{F}_Z(x, y) = -\langle y, \overline{Z}\rangle$, where $\overline{Z} = \frac{1}{n}\sum_{i=1}^n z_i$, $\widehat{y}_Z^*(x) = \zeta x$, and $\widehat{\Phi}_Z(x) = \widehat{F}_Z(x, \widehat{y}_Z^*(x)) = \langle -\zeta x, \overline{Z}\rangle \implies \widehat{x}^*(Z) = \arg\min_{x \in \mathcal{X}} \widehat{\Phi}_Z(x) = \frac{\overline{Z}}{\|\overline{Z}\|}$. Therefore, for any $x \in \mathcal{X}$, we have

$$\widehat{\Phi}_Z(x) - \widehat{\Phi}_Z(\widehat{x}^*(Z)) = -\zeta\left\langle \overline{Z}, x - \frac{\overline{Z}}{\|\overline{Z}\|}\right\rangle$$

$$= \zeta \left[ \|\overline{Z}\| \left( 1 - \langle x, \widehat{x}^*(Z) \rangle \right) \right]$$
$$\geq \frac{\zeta}{2} \left[ \|\overline{Z}\| \|x - \widehat{x}^*(Z)\|^2 \right], \tag{24}$$

since $\|x\|, \|\widehat{x}^*(Z)\| \leq 1$. Now, recall the following result, which is due to [9, Lemma 5.1] and [48, Theorem 1.1]:

**Lemma B.18** (Lower bounds for 1-way marginals). *Let $n, d \geq 1, \varepsilon > 0, 2^{-\Omega(n)} \leq \delta \leq 1/n^{1+\Omega(1)}$.*

1. *$\varepsilon$-DP algorithms: There is a number $M = \Omega(\min(n, d/\varepsilon))$ such that for every $\varepsilon$-DP $\mathcal{A}$, there is a data set $Z = (z_1, \ldots, z_n) \subset \{\pm 1/\sqrt{d}\}^d$ with $\|\overline{Z}\| \in [(M-1)/n, (M+1)/n]$ such that, with probability at least $1/2$ over the algorithm random coins, we have*
$$\|\mathcal{A}(Z) - \overline{Z}\| = \Omega\left( \min\left( 1, \frac{d}{\varepsilon n} \right) \right).$$

2. *$(\varepsilon, \delta)$-DP algorithms: There is a number $M = \Omega(\min(n, \sqrt{d \log(1/\delta)}/\varepsilon))$ such that for every $(\varepsilon, \delta)$-DP $\mathcal{A}$, there is a data set $Z = (z_1, \ldots, z_n) \subset \{\pm 1/\sqrt{d}\}^d$ with $\|\overline{Z}\| \in [(M-1)/n, (M+1)/n]$ such that, with probability at least $1/3$ over the algorithm random coins, we have*
$$\|\mathcal{A}(Z) - \overline{Z}\| = \Omega\left( \min\left( 1, \frac{\sqrt{d \log(1/\delta)}}{\varepsilon n} \right) \right).$$

We claim there exists $Z \in \mathcal{Z}^n$ with $\|\overline{Z}\| \in [(M-1)/n, (M+1)/n]$ such that
$$\left\| \mathcal{A}(Z) - \frac{\overline{Z}}{\|\overline{Z}\|} \right\| \gtrsim 1 \tag{25}$$

with probability at least $1/2$. Suppose for the sake of contradiction that $\forall Z \in \mathcal{Z}^n$ with $\|\overline{Z}\| \in [(M-1)/n, (M+1)/n]$, we have
$$\left\| \mathcal{A}(Z) - \frac{\overline{Z}}{\|\overline{Z}\|} \right\| \ll 1$$

with probability at least $1/2$. Let $c \in [-1/n, 1/n]$ such that $\|\overline{Z}\| = M/n + c$. Then for the $\varepsilon$-DP algorithm $\tilde{\mathcal{A}}(Z) := \frac{M}{n} \mathcal{A}(Z)$, we have

$$\|\tilde{\mathcal{A}}(Z) - \overline{Z}\| = \left\| \frac{M}{n} \mathcal{A}(Z) - \overline{Z} \right\|$$
$$= \left\| \frac{M}{n} \mathcal{A}(Z) - \left( \frac{M}{n} + c \right) \frac{\overline{Z}}{\|\overline{Z}\|} \right\|$$
$$\leq \left\| \frac{M}{n} \left[ \mathcal{A}(Z) - \frac{\overline{Z}}{\|\overline{Z}\|} \right] \right\| + c \left\| \frac{\overline{Z}}{\|\overline{Z}\|} \right\|$$
$$\ll \frac{M}{n} + c \leq \frac{M+1}{n},$$

which implies $\|\tilde{\mathcal{A}}(Z) - \overline{Z}\| \ll 1 \wedge \frac{d}{\varepsilon n}$, contradicting Lemma B.18. By combining the claim (25) with inequality (24), we conclude that if $x = \mathcal{A}(Z)$ is $\varepsilon$-DP, then

$$\widehat{\Phi}_Z(x) - \widehat{\Phi}_Z^* \geq \frac{\zeta}{2} \left[ \|\overline{Z}\| \|x - \widehat{x}^*(Z)\|^2 \right]$$
$$\gtrsim \zeta \frac{M}{n} \cdot 1 \gtrsim \zeta \min\left\{ 1, \frac{d}{n\varepsilon} \right\}.$$

Next, we scale our hard instance to obtain the $\varepsilon$-DP lower bound. Define the scaled parameter domains $\tilde{X} = D_x \mathbb{B}, \tilde{Y} = D_y \mathbb{B}, \tilde{Z} = \mathcal{Z} = \{\pm 1/\sqrt{d}\}^d$, and denote $\tilde{x} = D_x x, \tilde{y} = D_y y$ for any $x, y \in \mathcal{X} \times \mathcal{Y} = \mathbb{B}^2$. Define $\tilde{f} : \tilde{X} \times \tilde{Y} \times \tilde{Z} \to \mathbb{R}$ by
$$\tilde{f}(\tilde{x}, \tilde{y}, \tilde{z}) = -L_{f,y} \langle \tilde{y}, \tilde{z} \rangle,$$

which is convex and $L_{f,y}$-Lipschitz in $y$ for any permissible $\tilde{x}, \tilde{z}$. Define $\tilde{g} : \tilde{X} \times \tilde{Y} \times \tilde{Z} \to \mathbb{R}$ by

$$\tilde{g}(\tilde{x}, \tilde{y}, \tilde{z}) = \frac{\mu_g}{2} \|\tilde{y} - \zeta\tilde{x}\|^2,$$

where

$$\zeta := D_y/D_x.$$

Then $\tilde{g}$ is $\mu_g$-strongly convex in $y$ and $2L_{g,y}$-Lipschitz, since $L_{g,y} \geq \mu_g D_y$. Now,

$$\tilde{F}_{\tilde{Z}}(\tilde{x}, \tilde{y}) := \frac{1}{n}\sum_{i=1}^{n} \tilde{f}(\tilde{x}, \tilde{y}, \tilde{z}_i) = -L_{f,y}\langle \tilde{y}, \overline{\tilde{Z}}\rangle,$$

$$\tilde{y}_{\tilde{Z}}^*(\tilde{x}) := \operatorname{argmin}_{\tilde{y}\in\mathbb{R}^{d_y}} \left[ \tilde{G}_{\tilde{Z}}(\tilde{x}, \tilde{y}) = \frac{\mu_g}{2}\|\tilde{y} - \zeta\tilde{x}\|^2 \right] = \zeta\tilde{x} \in \tilde{\mathcal{Y}},$$

and

$$\tilde{\Phi}(\tilde{x}) := \tilde{F}_{\tilde{Z}}(\tilde{x}, \tilde{y}_{\tilde{Z}}^*(\tilde{x})) = -L_{f,y}\langle \zeta\tilde{x}, \overline{\tilde{Z}}\rangle.$$

Also,

$$\tilde{x}^*(\tilde{Z}) := \operatorname{argmin}_{\tilde{x}\in\tilde{\mathcal{X}}} \tilde{\Phi}(\tilde{x}) = \frac{\overline{\tilde{Z}}}{\|\overline{\tilde{Z}}\|} D_x = D_x \hat{x}^*(Z) = D_x \frac{\overline{Z}}{\|\overline{Z}\|}.$$

Thus, for any $\varepsilon$-DP $\mathcal{A}$, there exists a dataset $Z = \tilde{Z}$ such that the following holds with probability at least $1/2$, where we denote $\tilde{x} = \mathcal{A}(\tilde{Z})$:

$$\tilde{\Phi}(\tilde{x}) - \tilde{\Phi}(\tilde{x}^*(\tilde{Z})) = -L_{f,y}\left[\zeta\langle\tilde{x}, \overline{\tilde{Z}}\rangle - \zeta\langle\tilde{x}_{\tilde{Z}}^*, \overline{\tilde{Z}}\rangle\right]$$

$$= -L_{f,y}\zeta\left[D_x\langle x - \hat{x}^*(Z), \overline{Z}\rangle\right]$$

$$= -L_{f,y}\zeta D_x\left\langle x - \frac{\overline{Z}}{\|\overline{Z}\|}, \overline{Z}\right\rangle$$

$$\geq \frac{L_{f,y}D_x\zeta}{2}\left[\|\overline{Z}\|\|x - \hat{x}^*(Z)\|\right]$$

$$\gtrsim L_{f,y}D_x\zeta\left[\frac{d}{\varepsilon n} \wedge 1\right]$$

$$= L_{f,y}D_y\left[\frac{d}{\varepsilon n} \wedge 1\right].$$

The argument for the $(\varepsilon, \delta)$-DP case is identical to the above, except we invoke part 2 of Lemma B.18 instead of part 1.

Finally, it is easy to verify that Assumptions 2.1 and 2.2 are satisfied, with $\beta_{g,xy} \leq \frac{\mu_g D_y}{D_x}$, $C_{g,xy} = C_{g,yy} = M_{g,yy} = M_{g,xy} = 0 = \beta_{f,xx} = \beta_{f,xy} = \beta_{f,yy}$.

**Case 2:** $L_{f,y}D_y \lesssim L_{f,x}D_x$. In this case, the desired lower bounds follow from a trivial reduction to the single-level DP ERM lower bounds of [9, Theorems 5.2 and 5.3]: take $\mathcal{Y} = \{y_0\}$ for some $y_0 \in \mathbb{R}^d$ with $\|y_0\| \leq D_y$, $\mathcal{X} = D_x\mathbb{B}$, $\mathcal{Z} = \{\pm 1/\sqrt{d}\}^d$, and let $f(x, y, z) = -L_{f,x}\langle x, z\rangle$ and $g(x, y, z) = \frac{\mu_g}{2}\|y\|^2$. Then $f$ and $g$ satisfy Assumption 2.1, $\hat{y}_Z^*(x) = y_0$, $\hat{F}_Z(x) = \hat{\Phi}_Z(x) = -L_{f,x}\langle x, \overline{Z}\rangle$. Thus, the lower bounds on the excess risk $\hat{F}_Z(x) - \hat{F}_Z^*$ for DP $x$ given in [9, Theorems 5.2 and 5.3] apply verbatim to the excess risk $\hat{\Phi}_Z(x) - \hat{\Phi}_Z^*$. This completes the proof. $\square$

# C Proofs for Section 4

## C.1 An iterative second-order method

We have the following key lemma, which will be needed for proving Theorem 4.2.

**Lemma C.1** (Re-statement of Lemma 4.1). *For any fixed $x_t$, define the query $q_t : \mathcal{Z}^n \to \mathbb{R}^d$,*

$$q_t(Z) := \overline{\nabla}\hat{F}_Z(x_t, y_{t+1}),$$

*where $y_{t+1} = y_{t+1}(Z)$ is given in Algorithm 1. If $\alpha \leq \frac{K}{Cn}$ where $C$ and $K$ are defined in Equations (11) and (12), then the $\ell_2$-sensitivity of $q_t$ is upper bounded by $\frac{4K}{n}$.*

*Proof.* We will need the following bound due to [23, Lemma 2.2]: for any $x, y \in \mathcal{X} \times \mathcal{Y}$,

$$\|\nabla\widehat{\Phi}_Z(x) - \overline{\nabla}\widehat{F}_Z(x, y)\| \leq C\|\widehat{y}_Z^*(x) - y\| \tag{26}$$

for

$$C = \beta_{f,xy} + \frac{\beta_{f,yy}\beta_{g,xy}}{\mu_g} + L_{f,y}\left(\frac{C_{g,xy}}{\mu_g} + \frac{C_{g,yy}\beta_{g,xy}}{\mu_g^2}\right).$$

Now, denoting $y_{t+1} = y_{t+1}(Z)$ and $y'_{t+1} = y_{t+1}(Z')$, the sensitivity of the the query $q_t$ is bounded by

$$\sup_{Z \sim Z'} \|q_t(Z) - q_t(Z')\|$$

$$= \sup_{Z \sim Z'} \|\overline{\nabla}\widehat{F}_Z(x_t, y_{t+1}) - \overline{\nabla}\widehat{F}_{Z'}(x_t, y'_{t+1})\|$$

$$\leq \sup_{Z \sim Z'} \left[\|\overline{\nabla}\widehat{F}_Z(x_t, y_{t+1}) - \nabla\widehat{\Phi}_Z(x_t)\| + \|\nabla\widehat{\Phi}_Z(x_t) - \nabla\widehat{\Phi}_{Z'}(x_t)\| + \|\nabla\widehat{\Phi}_{Z'}(x_t) - \overline{\nabla}\widehat{F}_{Z'}(x_t, y'_{t+1})\|\right]$$

$$\leq C\|y_{t+1} - \widehat{y}_Z^*(x_t)\| + \|\nabla\widehat{\Phi}_Z(x_t) - \nabla\widehat{\Phi}_{Z'}(x_t)\| + C\|y'_{t+1} - \widehat{y}_{Z'}^*(x_t)\|$$

$$\leq 2C\alpha + \|\nabla\widehat{\Phi}_Z(x_t) - \nabla\widehat{\Phi}_{Z'}(x_t)\|$$

$$\leq \frac{2K}{n} + \|\nabla\widehat{\Phi}_Z(x_t) - \nabla\widehat{\Phi}_{Z'}(x_t)\|,$$

where we used the bound (26) and our choice of $\alpha$, for $K$ defined in the theorem statement. Next, we claim

$$\|\nabla\widehat{\Phi}_Z(x_t) - \nabla\widehat{\Phi}_{Z'}(x_t)\| \leq \frac{2K}{n}. \tag{27}$$

This will follow from a rather long calculation that uses Assumption 2.2 repeatedly, along with the perturbation inequality $\|M^{-1} - N^{-1}\| \leq \|M^{-1}\|\|M - N\|\|N^{-1}\|$ which holds for any invertible matrices $M$ and $N$. Let us now prove the bound (27). In what follows, the notation $\nabla$ denote the derivative of the function w.r.t. $x$ (accounting for the dependence of the function on $\widehat{y}_Z^*(x)$ via the chain rule) and denote

$$M_Z(x, y) := \nabla_{xy}^2\widehat{G}_Z(x, y)[\nabla_{yy}^2\widehat{G}_Z(x, y)]^{-1}.$$

Then,

$$\|\nabla\widehat{\Phi}_Z(x_t) - \nabla\widehat{\Phi}_{Z'}(x_t)\|$$

$$\leq \frac{1}{n}\left\|\sum_{i=1}^n \nabla f(x, \widehat{y}_Z^*(x), z_i) - \nabla f(x, \widehat{y}_{Z'}^*(x), z_i)\right\| + \frac{1}{n}\left\|\sum_{i=1}^n \nabla f(x, \widehat{y}_{Z'}^*(x), z_i) - \nabla f(x, \widehat{y}_{Z'}^*(x), z_i')\right\|$$

$$\leq \frac{1}{n}\sum_{i=1}^n \|\nabla_x f(x, \widehat{y}_Z^*(x), z_i) - \nabla_x f(x, \widehat{y}_{Z'}^*(x), z_i)\|$$

$$+ \frac{1}{n}\sum_{i=1}^n \|M_{Z'}(x, \widehat{y}_{Z'}^*(x))\nabla_y f(x, \widehat{y}_{Z'}^*(x), z_i) - M_Z(x, \widehat{y}_Z^*(x))\nabla_y f(x, \widehat{y}_Z^*(x), z_i)\|$$

$$+ \frac{1}{n}\left\|\sum_{i=1}^n \nabla f(x, \widehat{y}_{Z'}^*(x), z_i) - \nabla f(x, \widehat{y}_{Z'}^*(x), z_i')\right\|$$

$$\leq \frac{1}{n}\sum_{i=1}^n \beta_{f,xy}\|\widehat{y}_Z^*(x) - \widehat{y}_{Z'}^*(x)\|$$

$$+ \frac{1}{n}\sum_{i=1}^n \|M_{Z'}(x, \widehat{y}_{Z'}^*(x))\nabla_y f(x, \widehat{y}_{Z'}^*(x), z_i) - M_Z(x, \widehat{y}_Z^*(x))\nabla_y f(x, \widehat{y}_Z^*(x), z_i)\|$$

$$+ \frac{1}{n}\|\nabla f(x, \widehat{y}_{Z'}^*(x), z_1) - \nabla f(x, \widehat{y}_{Z'}^*(x), z_1')\|,$$

where we assumed WLOG that $z_1 \neq z_1'$ and used the smoothness assumption in the last inequality above. Now, recall that

$$\|\widehat{y}_Z^*(x) - \widehat{y}_{Z'}^*(x)\| \leq \frac{2L_{g,y}}{\mu_g n}$$

and note that

$$\|\nabla f(x, \widehat{y}^*_{Z'}(x), z_1) - \nabla f(x, \widehat{y}^*_{Z'}(x), z_1')\| \leq 2\overline{L} = 2\left(L_{f,x} + \frac{L_{f,y}\beta_{g,xy}}{\mu_g}\right),$$

by the chain rule and $(\beta_{g,xy}/\mu_g)$-Lipschitz continuity of $\widehat{y}^*_Z$ (see [23] for a proof of this result). Thus,

$$\|\nabla\widehat{\Phi}_Z(x_t) - \nabla\widehat{\Phi}_{Z'}(x_t)\| \leq \frac{1}{n}\sum_{i=1}^n \beta_{f,xy}\frac{2L_{g,y}}{\mu_g n}$$

$$+ \frac{1}{n}\sum_{i=1}^n \|M_{Z'}(x, \widehat{y}^*_{Z'}(x))\nabla_y f(x, \widehat{y}^*_{Z'}(x), z_i) - M_Z(x, \widehat{y}^*_Z(x))\nabla_y f(x, \widehat{y}^*_Z(x), z_i)\|$$

$$+ \frac{2\overline{L}}{n}.$$

Next, we bound

$$\|M_{Z'}(x, \widehat{y}^*_{Z'}(x))\nabla_y f(x, \widehat{y}^*_{Z'}(x), z_i) - M_Z(x, \widehat{y}^*_Z(x))\nabla_y f(x, \widehat{y}^*_Z(x), z_i)\|$$

$$\leq \sup_{x,Z}\left[\|M_Z(x, \widehat{y}^*_Z(x))\|\right]\|\nabla_y f(x, \widehat{y}^*_Z(x), z_i) - \nabla_y f(x, \widehat{y}^*_{Z'}(x), z_i)\|$$

$$+ \sup_{x,Z}\left[\|\nabla_y f(x, \widehat{y}^*_Z(x), z_i)\|\right]\|M_Z(x, \widehat{y}^*_Z(x)) - M_{Z'}(x, \widehat{y}^*_{Z'}(x))\|$$

$$\leq \frac{\beta_{g,xy}}{\mu_g}\beta_{f,yy}\frac{2L_{g,y}}{\mu_g n} + L_{f,y}\|M_Z(x, \widehat{y}^*_Z(x)) - M_{Z'}(x, \widehat{y}^*_{Z'}(x))\|.$$

It remains to bound

$$\|M_Z(x, \widehat{y}^*_Z(x)) - M_{Z'}(x, \widehat{y}^*_{Z'}(x))\|$$

$$\leq \left\|\nabla^2_{xy}\widehat{G}_Z(x, \widehat{y}^*_Z(x))\nabla^2_{yy}\widehat{G}_Z(x, \widehat{y}^*_Z(x))^{-1} - \nabla^2_{xy}\widehat{G}_Z(x, \widehat{y}^*_{Z'}(x))\nabla^2_{yy}\widehat{G}_Z(x, \widehat{y}^*_Z(x))^{-1}\right\|$$

$$+ \left\|\nabla^2_{xy}\widehat{G}_Z(x, \widehat{y}^*_{Z'}(x))\nabla^2_{yy}\widehat{G}_Z(x, \widehat{y}^*_Z(x))^{-1} - \nabla^2_{xy}\widehat{G}_Z(x, \widehat{y}^*_{Z'}(x))\nabla^2_{yy}\widehat{G}_Z(x, \widehat{y}^*_{Z'}(x))^{-1}\right\|$$

$$+ \left\|\nabla^2_{xy}\widehat{G}_{Z'}(x, \widehat{y}^*_{Z'}(x))\nabla^2_{yy}\widehat{G}_{Z'}(x, \widehat{y}^*_{Z'}(x))^{-1} - \nabla^2_{xy}\widehat{G}_{Z'}(x, \widehat{y}^*_{Z'}(x))\nabla^2_{yy}\widehat{G}_Z(x, \widehat{y}^*_{Z'}(x))^{-1}\right\|$$

$$+ \left\|\nabla^2_{xy}\widehat{G}_{Z'}(x, \widehat{y}^*_{Z'}(x))\nabla^2_{yy}\widehat{G}_Z(x, \widehat{y}^*_{Z'}(x))^{-1} - \nabla^2_{xy}\widehat{G}_Z(x, \widehat{y}^*_{Z'}(x))\nabla^2_{yy}\widehat{G}_Z(x, \widehat{y}^*_{Z'}(x))^{-1}\right\|$$

$$\leq \frac{C_{g,xy}\|\widehat{y}^*_Z(x) - \widehat{y}^*_{Z'}(x)\|}{\mu_g}$$

$$+ \beta_{g,xy}\left\|\nabla^2_{yy}\widehat{G}_Z(x, \widehat{y}^*_Z(x))^{-1} - \nabla^2_{yy}\widehat{G}_Z(x, \widehat{y}^*_{Z'}(x))^{-1}\right\|$$

$$+ \beta_{g,xy}\left\|\nabla^2_{yy}\widehat{G}_{Z'}(x, \widehat{y}^*_{Z'}(x))^{-1} - \nabla^2_{yy}\widehat{G}_Z(x, \widehat{y}^*_{Z'}(x))^{-1}\right\|$$

$$+ \frac{2\beta_{g,xy}}{\mu_g n}$$

$$\leq \frac{2C_{g,xy}L_{g,y}}{\mu_g^2 n}$$

$$+ \beta_{g,xy}\left\|\nabla^2_{yy}\widehat{G}_Z(x, \widehat{y}^*_Z(x))^{-1} - \nabla^2_{yy}\widehat{G}_Z(x, \widehat{y}^*_{Z'}(x))^{-1}\right\|$$

$$+ \beta_{g,xy}\left\|\nabla^2_{yy}\widehat{G}_{Z'}(x, \widehat{y}^*_{Z'}(x))^{-1} - \nabla^2_{yy}\widehat{G}_Z(x, \widehat{y}^*_{Z'}(x))^{-1}\right\|$$

$$+ \frac{2\beta_{g,xy}}{\mu_g n}$$

$$\leq \frac{2C_{g,xy}L_{g,y}}{\mu_g^2 n}$$

$$+ \beta_{g,xy}\frac{C_{g,yy}\|\widehat{y}^*_Z(x) - \widehat{y}^*_{Z'}(x)\|}{\mu_g^2}$$

$$+ \beta_{g,xy}\frac{2\beta_{g,yy}}{\mu_g^2 n}$$

$$+ \frac{2\beta_{g,xy}}{\mu_g n}$$

$$\leq \frac{2C_{g,xy}L_{g,y}}{\mu_g^2 n}$$

$$+ \beta_{g,xy}\frac{2C_{g,yy}L_{g,y}}{\mu_g^3 n}$$

$$+ \beta_{g,xy}\frac{2\beta_{g,yy}}{\mu_g^2 n}$$

$$+ \frac{2\beta_{g,xy}}{\mu_g n},$$

where in the second-to-last inequality we used the operator norm inequality

$$\|M^{-1} - N^{-1}\| \leq \|M^{-1}\|\|M - N\|\|N^{-1}\|,$$

which holds for any invertible matrices $M$ and $N$ of compatible shape.

Combining the above pieces completes the proof. $\qquad\square$

We have the following refinement of [23, Lemma 2.2c], in which we correctly describe the precise dependence on the smoothness, Lipschitz, and strong convexity parameters of $f$ and $g$:

**Lemma C.2** (Smoothness of $\widehat{\Phi}_Z$)**.** *Grant Assumptions 2.1 and 2.2. Then, for any $x_1, x_2$,*

$$\|\nabla\widehat{\Phi}_Z(x_1) - \nabla\widehat{\Phi}_Z(x_2)\| \leq \beta_\Phi \|x_1 - x_2\|,$$

*where*

$$\beta_\Phi := \beta_{f,xx} + \frac{2\beta_{f,xy}\beta_{g,xy}}{\mu_g} + \frac{\beta_{g,xy}^2\beta_{f,yy}}{\mu_g^2} + \frac{L_{f,y}\beta_{g,xy}}{\mu_g^2}\left(M_{g,yy} + \frac{C_{g,yy}\beta_{g,xy}}{\mu_g}\right) + \frac{L_{f,y}C_{g,xy}\beta_{g,xy}}{\mu_g^2} + \frac{L_{f,y}M_{g,xy}}{\mu_g}.$$

$$\tag{28}$$

*Proof.* Recall that

$$\nabla\widehat{\Phi}_Z(x) = \nabla_x\widehat{F}_Z(x, \widehat{y}_Z^*(x)) - M(x, \widehat{y}_Z^*(x))\nabla_y\widehat{F}_Z(x, \widehat{y}_Z^*(x)),$$

where

$$M(x, y) := \nabla_{xy}^2\widehat{G}_Z(x, y)[\nabla_{yy}^2\widehat{G}_Z(x, y)]^{-1}.$$

Also, $\widehat{y}_Z^*$ is $\frac{\beta_{g,xy}}{\mu_g}$-Lipschitz (c.f. [23, Lemma 2.2b]). Therefore,

$$\|\nabla\widehat{\Phi}_Z(x_1) - \nabla\widehat{\Phi}_Z(x_2)\| \leq \|\nabla_x\widehat{F}_Z(x_1, \widehat{y}_Z^*(x_1)) - \nabla_x\widehat{\Phi}_Z(x_2, \widehat{y}_Z^*(x_2))\|$$

$$+ \|M(x_1, \widehat{y}_Z^*(x_1))\nabla_y\widehat{F}_Z(x_1, \widehat{y}_Z^*(x_1)) - M(x_2, \widehat{y}_Z^*(x_2))\nabla_y\widehat{F}_Z(x_2, \widehat{y}_Z^*(x_2))\|$$

$$\leq \beta_{f,xx}\|x_1 - x_2\| + \beta_{f,xy}\|\widehat{y}_Z^*(x_1) - \widehat{y}_Z^*(x_2)\|$$

$$+ \|M(x_1, \widehat{y}_Z^*(x_1))\|\|\nabla_y\widehat{F}_Z(x_1, \widehat{y}_Z^*(x_1)) - \widehat{F}_Z(x_2, \widehat{y}_Z^*(x_2))\| + \|\nabla_y\widehat{F}_Z(x_2, \widehat{y}_Z^*(x_2))\|\|M(x_1, \widehat{y}_Z^*(x_1)) - M(x_2, \widehat{y}_Z^*(x_2))\|$$

$$\leq \left(\beta_{f,xx} + \frac{\beta_{f,xy}\beta_{g,xy}}{\mu_g}\right)\|x_1 - x_2\| + \frac{\beta_{g,xy}}{\mu_g}\|\nabla_y\widehat{F}_Z(x_1, \widehat{y}_Z^*(x_1)) - \widehat{F}_Z(x_2, \widehat{y}_Z^*(x_2))\|$$

$$+ L_{f,y}\left\|\nabla_{xy}^2\widehat{G}_Z(x_1, \widehat{y}_Z^*(x_1))\right\|\left\|\left[\nabla_{yy}^2\widehat{G}_Z(x_1, \widehat{y}_Z^*(x_1)]\right]^{-1} - \left[\nabla_{yy}^2\widehat{G}_Z(x_2, \widehat{y}_Z^*(x_2))\right]^{-1}\right\|$$

$$+ L_{f,y}\left\|\left[\nabla_{yy}^2\widehat{G}_Z(x_2, \widehat{y}_Z^*(x_2))\right]^{-1}\right\|\left\|\nabla_{xy}^2\widehat{G}_Z(x_1, \widehat{y}_Z^*(x_1)) - \nabla_{xy}^2\widehat{G}_Z(x_2, \widehat{y}_Z^*(x_2))\right\|$$

$$\leq \left(\beta_{f,xx} + \frac{\beta_{f,xy}\beta_{g,xy}}{\mu_g}\right)\|x_1 - x_2\| + \frac{\beta_{g,xy}}{\mu_g}\left(\beta_{f,yy}\frac{\beta_{g,xy}}{\mu_g}\|x_1 - x_2\| + \beta_{f,xy}\|x_1 - x_2\|\right)$$

$$+ L_{f,y}\beta_{g,xy}\left\|\nabla_{yy}^2\widehat{G}_Z(x_1, \widehat{y}_Z^*(x_1))^{-1} - \widehat{G}_Z(x_2, \widehat{y}_Z^*(x_2))^{-1}\right\| + \frac{L_{f,y}}{\mu_g}\left\|\nabla_{xy}^2\widehat{G}_Z(x_1, \widehat{y}_Z^*(x_1)) - \nabla_{xy}^2\widehat{G}_Z(x_2, \widehat{y}_Z^*(x_2))\right\|$$

$$\leq \left(\beta_{f,xx} + \frac{2\beta_{f,xy}\beta_{g,xy}}{\mu_g} + \frac{\beta_{g,xy}^2 \beta_{f,yy}}{\mu_g^2}\right) \|x_1 - x_2\|$$

$$+ \frac{L_{f,y}\beta_{g,xy}}{\mu_g^2} \left\|\nabla_{yy}^2 \widehat{G}_Z(x_1, \widehat{y}_Z^*(x_1)) - \nabla_{yy}^2 \widehat{G}_Z(x_2, \widehat{y}_Z^*(x_2))\right\| + \frac{L_{f,y}}{\mu_g} \left(C_{g,xy}\|\widehat{y}_Z^*(x_1) - \widehat{y}_Z^*(x_2)\| + M_{g,xy}\|x_1 - x_2\|\right)$$

$$\leq \left(\beta_{f,xx} + \frac{2\beta_{f,xy}\beta_{g,xy}}{\mu_g} + \frac{\beta_{g,xy}^2 \beta_{f,yy}}{\mu_g^2}\right) \|x_1 - x_2\|$$

$$+ \frac{L_{f,y}\beta_{g,xy}}{\mu_g^2} \left[M_{g,yy}\|x_1 - x_2\| + C_{g,yy}\|\widehat{y}_Z^*(x_1) - \widehat{y}_Z^*(x_2)\|\right] + \frac{L_{f,y}}{\mu_g}\left(C_{g,xy}\frac{\beta_{g,xy}}{\mu_g}\|x_1 - x_2\| + M_{g,xy}\|x_1 - x_2\|\right),$$

where we used the operator norm inequality

$$\|M^{-1} - N^{-1}\| \leq \|M^{-1}\|\|M - N\|\|N^{-1}\|,$$

which holds for any invertible matrices $M$ and $N$ of compatible shape. Using the Lipschitz continuity of $\widehat{y}_Z^*$ one last time completes the proof. □

**Theorem C.3** (Precise version of Theorem 4.2). *Grant Assumptions 2.1 and 2.2. Set $\sigma = 32K\sqrt{T\log(1/\delta)}/n\varepsilon$ and*

$$\alpha = \min\left(\frac{K}{nC}, \frac{1}{C}\left[K\sqrt{\left(\widehat{\Phi}_Z(x_0) - \widehat{\Phi}_Z^*\right)\beta_\Phi}\frac{\sqrt{d_x \log(1/\delta)}}{\varepsilon n}\right]^{1/2}\right)$$

*for $C$ defined in Equation (11) in Algorithm 1, where*

$$K = 2\left[\frac{\beta_{f,xy}L_{g,y}}{\mu_g} + 2\overline{L} + \frac{\beta_{g,xy}\beta_{f,yy}L_{g,y}}{\mu_g^2} + \frac{L_{f,y}C_{g,xy}L_{g,y}}{\mu_g^2} + \frac{L_{f,y}\beta_{g,xy}L_{g,y}C_{g,yy}}{\mu_g^3} + \frac{L_{f,y}\beta_{g,yy}\beta_{g,xy}}{\mu_g^2}\right].$$

*Then, Algorithm 1 is $(\varepsilon, \delta)$-DP. Further, choosing $\eta = 1/2\beta_\Phi$ and $T = \left\lceil \frac{n\varepsilon}{\sqrt{d_x \log(1/\delta)}}\frac{\sqrt{\beta_\Phi(\widehat{\Phi}_Z(x_0) - \widehat{\Phi}_Z^*)}}{K}\right\rceil$ for $\beta_\Phi$ defined in Equation (28), the output of Algorithm 1 satisfies*

$$\mathbb{E}\|\nabla\widehat{\Phi}_Z(\widehat{x}_T)\| \lesssim \left[K\sqrt{\left(\widehat{\Phi}_Z(x_0) - \widehat{\Phi}_Z^*\right)\beta_\Phi}\frac{\sqrt{d_x \log(1/\delta)}}{\varepsilon n}\right]^{1/2}.$$

*Proof.* **Privacy:** By Lemma 4.1, the $\ell_2$-sensitivity of $\overline{\nabla}\widehat{F}_Z(x_t, y_{t+1})$ is upper bounded by $4K/n$. Thus, by the privacy guarantee of the gaussian mechanism and the advanced composition theorem, our prescribed choice of $\sigma$ ensures that all $T$ iterations of Algorithm 1 satisfy $(\varepsilon, \delta)$-DP. Hence $\widehat{x}_T$ is $(\varepsilon, \delta)$-DP by post-processing.

**Utility:** We will need the following descent lemma for gradient descent with biased, noisy gradient oracle:

**Lemma C.4.** *[1, Lemma 2] Let $H$ be $\beta$-smooth, $x_{t+1} = x_t - \eta\tilde{\nabla}H(x_t)$, where $\tilde{\nabla}H(x_t) = \nabla H(x_t) + b_t + N_t$ is a biased, noisy gradient such that $\mathbb{E}[N_t|x_t] = 0$, $\|\mathbb{E}[b_t|x_t]\| \leq B$, and $\mathbb{E}\left[\|N_t\|^2|x_t\right] \leq \Sigma^2$. Then for any stepsize $\eta \leq \frac{1}{2\beta}$, we have*

$$\mathbb{E}[H(x_{t+1}) - H(x_t)|x_t] \leq -\frac{\eta}{2}\|\nabla H(x_t)\|^2 + \frac{\eta}{2}B^2 + \frac{\eta^2\beta}{2}\Sigma^2. \tag{29}$$

We will apply Lemma C.4 to $H = \widehat{\Phi}_Z$ which is $\beta_\Phi$-smooth by Lemma C.2, $\tilde{\nabla}H(x_t) = \overline{\nabla}\widehat{F}_Z(x_t, y_{t+1}) + u_t$ with bias $b_t = \overline{\nabla}\widehat{F}_Z(x_t, y_{t+1}) - \nabla\widehat{\Phi}_Z(x_t)$ and noise $N_t = u_t$:

$$\mathbb{E}[\widehat{\Phi}_Z(x_{t+1}) - \widehat{\Phi}_Z(x_t)|x_t] \leq -\frac{\eta}{2}\|\nabla\widehat{\Phi}_Z(x_t)\|^2 + \frac{\eta}{2}B^2 + \frac{\eta^2\beta_\Phi}{2}\Sigma^2$$

$$\implies \mathbb{E}\|\nabla\widehat{\Phi}_Z(x_t)\|^2 \leq \frac{2}{\eta}\mathbb{E}[\widehat{\Phi}_Z(x_t) - \widehat{\Phi}_Z(x_{t+1})|x_t] + B^2 + \eta\beta_\Phi\Sigma^2$$

$$\implies \mathbb{E}\|\nabla\widehat{\Phi}_Z(\widehat{x}_T)\|^2 = \frac{1}{T}\sum_{t=1}^{T}\mathbb{E}\|\nabla\widehat{\Phi}_Z(x_t)\|^2 \leq \frac{2\left(\widehat{\Phi}_Z(x_0) - \widehat{\Phi}_Z^*\right)}{\eta T} + B^2 + \eta\beta_\Phi\Sigma^2 \qquad (30)$$

for any $\eta \leq \frac{1}{2\beta_\Phi}$.

Now, [23, Lemma 2.2a] tells us that

$$\|\overline{\nabla}\widehat{F}_Z(x_t, y_{t+1}) - \nabla\widehat{\Phi}_Z(x_t)\| \leq C\|\widehat{y}_Z^*(x_t) - y_{t+1}\|,$$

for $C$ defined in Equation (11). Therefore,

$$B = \|\mathbb{E}[b_t|x_t]\| \leq C\alpha \leq \left[K\sqrt{\left(\widehat{\Phi}_Z(x_0) - \widehat{\Phi}_Z^*\right)\beta_\Phi}\frac{\sqrt{d_x\log(1/\delta)}}{\varepsilon n}\right]^{1/2}$$

by our choice of $\alpha$. Further,

$$\Sigma^2 = \mathbb{E}\left[\|u_t\|^2|x_t\right] = d_x\sigma^2 = \frac{1024 d_x K^2 T\log(1/\delta)}{n^2\varepsilon^2}.$$

Plugging these values into (30) and choosing $\eta = 1/(2\beta_\Phi)$, we obtain

$$\mathbb{E}\|\nabla\widehat{\Phi}_Z(\widehat{x}_T)\|^2 \leq \frac{2\left(\widehat{\Phi}_Z(x_0) - \widehat{\Phi}_Z^*\right)}{\eta T} + B^2 + \eta\beta_\Phi\Sigma^2$$

$$\leq \frac{4\beta_\Phi\left(\widehat{\Phi}_Z(x_0) - \widehat{\Phi}_Z^*\right)}{T} + K\sqrt{\left(\widehat{\Phi}_Z(x_0) - \widehat{\Phi}_Z^*\right)\beta_\Phi}\frac{\sqrt{d_x\log(1/\delta)}}{\varepsilon n} + \frac{1024 d_x K^2 T\log(1/\delta)}{n^2\varepsilon^2}.$$

Plugging in the prescribed $T$ from the theorem statement and then using Jensen's inequality completes the proof.

$\square$

## C.2 "Warm starting" Algorithm 1 with the exponential mechanism

---
**Algorithm 2:** Warm-Start Meta-Algorithm for Bilevel ERM

---
1 **Input:** Data $Z \in \mathcal{Z}^n$, loss functions $f$ and $g$, privacy parameters $(\varepsilon, \delta)$, warm-start DP-ERM algorithm $\mathcal{A}$, DP-ERM stationary point finder $\mathcal{B}$;
2 Run $(\varepsilon/2, \delta/2)$-DP $\mathcal{A}$ on $\widehat{\Phi}_Z(\cdot)$ to obtain $x_0$;
3 Run $(\varepsilon/2, \delta/2)$-DP $\mathcal{B}$ on $\widehat{\Phi}_Z(\cdot)$ with initialization $x_0$ to obtain $x_{\text{priv}}$;
4 **Return:** $x_{\text{priv}}$.

---

We instantiate this framework by choosing $\mathcal{A}$ as the exponential mechanism (4) and $\mathcal{B}$ as Algorithm 1 to obtain the following result:

**Theorem C.5** (Re-statement of Theorem 4.3). *Grant Assumptions 2.1 and 2.2. Assume that there is a compact set $\mathcal{X} \subset \mathbb{R}^{d_x}$ of diameter $D_x$ containing an approximate global minimizer $\widehat{x}$ such that $\widehat{\Phi}_Z(\widehat{x}) - \widehat{\Phi}_Z^* \leq \Psi\frac{d}{\varepsilon n}$, where*

$$\Psi := L_{f,x}D_x + L_{f,y}D_y + \frac{L_{f,y}L_{g,y}}{\mu_g}.$$

*Then, there exists an $(\varepsilon, \delta)$-DP instantiation of Algorithm 2 with output satisfying*

$$\mathbb{E}\|\nabla\widehat{\Phi}_Z(x_{priv})\| \leq \tilde{O}\left(\left[K\Psi^{1/2}\beta_\Phi^{1/2}\right]^{1/2}\left(\frac{d_x\sqrt{\log(1/\delta)}}{(n\varepsilon)^{3/2}}\right)^{1/2}\right).$$

*Proof.* **Privacy:** This is immediate from basic composition, since $\mathcal{A}$ is $\varepsilon/2$-DP and $\mathcal{B}$ is $(\varepsilon/2, \delta/2)$-DP.

**Utility:** First, note that the output $x_0$ of the exponential mechanism in (4) satisfies

$$\widehat{\Phi}_Z(x_0) - \widehat{\Phi}_Z^* \leq \widetilde{O}\left(\frac{d_x}{\varepsilon n}\left[L_{f,x}D_x + L_{f,y}D_y + \frac{L_{f,y}L_{g,y}}{\mu_g}\right]\right)$$

with probability $\geq 1 - \zeta$ for any $\zeta > 0$ that is polynomial in all problem parameters, by [19, Theorem 3.11]. Let us say $x_0$ *is good* if the above excess risk bound holds. Now, by Theorem 4.2, the output of Algorithm 1 satisfies

$$\mathbb{E}\|\nabla\widehat{\Phi}_Z(\widehat{x}_T)\| \lesssim \left[K\sqrt{\left(\widehat{\Phi}_Z(x_0) - \widehat{\Phi}_Z^*\right)\beta_\Phi}\frac{\sqrt{d_x\log(1/\delta)}}{\varepsilon n}\right]^{1/2}.$$

Therefore,

$$\mathbb{E}\left[\|\nabla\widehat{\Phi}_Z(\widehat{x}_T)\|\big|x_0 \text{ is good}\right] \leq \tilde{O}\left(\left[K\sqrt{\left(\frac{d_x}{\varepsilon n}\left[L_{f,x}D_x + L_{f,y}D_y + \frac{L_{f,y}L_{g,y}}{\mu_g}\right]\right)\beta_\Phi}\frac{\sqrt{d_x\log(1/\delta)}}{\varepsilon n}\right]^{1/2}\right)$$

$$= \tilde{O}\left(\left[K\sqrt{\beta_\Phi\Psi\left(\frac{d_x}{\varepsilon n}\right)}\frac{\sqrt{d_x\log(1/\delta)}}{\varepsilon n}\right]^{1/2}\right).$$

Now, since $\widehat{\Phi}_Z(x) - \widehat{\Phi}_Z^* \leq \overline{L}D_x$ for any $x \in \mathcal{X}$, the law of total expectation implies

$$\mathbb{E}\left[\|\nabla\widehat{\Phi}_Z(\widehat{x}_T)\|\right] \leq \mathbb{E}\left[\|\nabla\widehat{\Phi}_Z(\widehat{x}_T)\|\big|x_0 \text{ is good}\right] + \overline{L}D_x\zeta$$

$$\leq \tilde{O}\left(\left[K\sqrt{\beta_\Phi\Psi\left(\frac{d_x}{\varepsilon n}\right)}\frac{\sqrt{d_x\log(1/\delta)}}{\varepsilon n}\right]^{1/2}\right) + \overline{L}D_x\zeta$$

$$\leq \tilde{O}\left(\left[K\sqrt{\beta_\Phi\Psi\left(\frac{d_x}{\varepsilon n}\right)}\frac{\sqrt{d_x\log(1/\delta)}}{\varepsilon n}\right]^{1/2}\right),$$

where the final inequality follows by choosing $\zeta$ sufficiently small. $\qquad\square$

## C.3 Deducing the upper bound in (3).

We prove in Lemma B.6 that $\sup_{Z \sim Z', x}\|\nabla\widehat{\Phi}_Z(x) - \nabla\widehat{\Phi}_{Z'}(x)\| \leq \frac{2G}{n}$, where $G$ is defined in (5). Thus, by similar arguments used to prove the results in Section 3.1, one can show that sampling $\widehat{x}$ proportional to the following density is $\varepsilon$-DP:

$$\propto \exp\left(-\frac{\varepsilon}{2G}\|\nabla\widehat{\Phi}_Z(\widehat{x})\|\right).$$

Moreover, the output of this sampler satisfies

$$\mathbb{E}\|\nabla\widehat{\Phi}_Z(\widehat{x})\| \leq O\left(G\frac{d_x}{\varepsilon n}\right). \tag{31}$$

Further, outputting arbitrary $x_0 \in \mathcal{X}$ trivially achieves $\|\nabla\widehat{\Phi}_Z(x_0)\| \leq L_{f,x}$ with 0-DP. By combining these upper bounds with our results in Theorems 4.2 and 4.3, we deduce the novel state-of-the-art upper bound in (3) for DP nonconvex bilevel ERM (with constant problem parameters).

## D  Limitations

While our work provides near-optimal rates and efficient algorithms for differentially private bilevel optimization (DP BLO), several limitations remain that should be considered when interpreting our theoretical and practical contributions.

**Assumptions on Problem Structure.**    Our results rely on several assumptions that may not hold in all practical settings. For convex DP BLO, we assume that the lower-level problem is strongly convex and that the loss functions are Lipschitz continuous with bounded gradients (and, for some of our algorithms, bounded and/or Lipschitz Hessians). These structural assumptions are standard in bilevel optimization theory but may not accurately capture real-world scenarios where lower-level problems are ill-conditioned, non-convex, or lack smoothness. Violations of these assumptions could degrade both utility and privacy guarantees, as our sensitivity and excess risk bounds depend critically on these properties.

**Scalability and Computational Efficiency.**    Although most of our algorithms are polynomial-time, they may still incur significant computational costs, especially in high-dimensional settings. Our efficient implementations rely on sampling techniques (e.g., grid-walk) whose runtime scales polynomially with the dimension. This may limit practicality on large-scale or high-dimensional problems. Additionally, the warm-start algorithm for nonconvex DP BLO is inefficient. We leave it for future work to develop algorithms with improved computational complexity guarantees.

**Lack of Empirical Validation.**    This paper focuses on theoretical analysis and does not include experimental results. While our theoretical rates are nearly optimal, empirical performance can depend on implementation details, constant factors, and practical optimization challenges not captured in our analysis. We defer empirical validation, including runtime measurements and real-data utility evaluation, to future work.

## E  Broader Impacts

This work advances algorithms for protecting the privacy of individuals whose data is used in bilevel learning applications, such as meta-learning and hyperparameter tuning. Privacy protection is widely regarded as a societal good and is enshrined as a fundamental right in many legal systems. By improving our theoretical understanding of privacy-preserving bilevel optimization, this work contributes to the development of machine learning methods that respect individual privacy.

However, there are trade-offs inherent in the use of differentially private (DP) methods. Privacy guarantees typically come at the cost of reduced model utility, which may lead to less accurate predictions or suboptimal decisions. For example, if a differentially private bilevel model is deployed in a sensitive application—such as medical treatment planning or environmental risk assessment—reduced accuracy could lead to unintended negative outcomes. While these risks are not unique to bilevel learning, they highlight the importance of transparency when communicating the limitations of DP models to stakeholders and decision-makers.

We also note that the performance of bilevel optimization algorithms depends on problem-specific factors such as the conditioning of the lower-level problem, the smoothness of the loss functions, and the dimensionality of the parameter spaces. Practitioners should carefully evaluate these factors when applying our methods in practice.

Finally, while this work focuses on theoretical developments and does not include empirical evaluation or deployment, we believe that the dissemination of privacy-preserving algorithms—alongside clear communication of their trade-offs—ultimately serves the public interest by empowering researchers and practitioners to build more responsible and privacy-aware machine learning systems.

