# OpenReview forum: "Differentially Private Bilevel Optimization: Efficient Algorithms with Near-Optimal Rates"
_NeurIPS.cc/2025/Conference — NeurIPS 2025 poster_

### Official Review · Reviewer_aszX · 2025-06-13

**Clarity:** 3
**Significance:** 3
**Originality:** 3
**Rating:** 5
**Confidence:** 2

**Summary:**

This work studies differentially private (DP) bilevel optimization (BLO) in both convex and non-convex settings. When the upper-level function is convex, the authors show that the (regularized) exponential mechanism (EM) achieves optimal excess empirical and generalization risks under (approximate and) pure DP. As EM can be computationally intensive, the authors further discuss about efficient implementations that allow for approximation errors. Back to theoretical analysis, they also establish excess risk lower bounds by constructing a hard instance. These bounds improve over existing results by an additive term that captures the complexity of the lower-level optimization problem. When the upper-level function is non-convex, the authors propose a second-order algorithm to avoid dimensionality dependence on the lower-level problem. Although the algorithm is computationally inefficient, as the authors themselves acknowledge, the theoretical advancement remains remarkable. Given the significant theoretical contributions to DP-BLO, I recommend acceptance.

**Questions:**

1. I noticed that the authors have discussed implementation-related issues, but I am still curious how practical it is to sampling from an exponential distribution like the one in Eq.(5), given the bilevel structure. Are there any readily available numerical packages/tools for this?
2. What is $D_y$? It first appears in line 150 without definition and is later used in many places. I assume that $D_y$ is intended to be the same as $Diam(\mathcal{Y})$ (or its upper bound) defined in line 113? Because this value appears many times in theorems and is directly related to your contributions, I suggest the authors clearly define it before first use.
3. (This point is more of a suggestion than a question.) You may consider moving the hard instance around line 692 constructed for proving the lower bound to some place immediate after Assumptions 2.1 & 2.2 as a motivating example to illustrate these assumptions. This may make it easier for readers to digest the inequalities in Assumptions 2.1, 2.2.

minor typos:
1. line 454: should be 'for all $\varepsilon>0$'
2. duplicate references [34] [35]

**Ethical Concerns:**

["NO or VERY MINOR ethics concerns only"]

**Final Justification:**

The authors have resolved my questions. I keep my original score.

However, after reading other reviews, I realize I might not fully understand some parts of the paper; in particular, the correlation issue mentioned by Reviewer pJMr. Therefore, I am lowering the confidence of my rating from 3 to 2.

**Limitations:**

The authors have discussed the limitations of this work in Appendix D, including computational challenges and lack of numerical experiments, on which I agree. I have no further comments on limitations.

**Quality:**

3

**Strengths And Weaknesses:**

Strengths:
1. This work provides a relatively comprehensive discussion about DP-BLO, covering a wide range of settings, including (non-)convex objectives, pure and approximate DP, empirical and generalization risks, etc. For each setting, algorithms with near optimal performance are proposed.
2. The authors establish improved lower bounds that capture the complexity of the lower-level problem.
3. Computational challenges are discussed

Weaknesses:
see Limitations section.

---

> ### Author Rebuttal · Authors · 2025-07-31
>
> Thank you for your thoughtful and detailed review. We respond to your comments below:
>
> >How practical it is to sampling from an exponential distribution like the one in Eq.(5), given the bilevel structure. Are there any readily available numerical packages/tools for this?
>
> While we show how to sample from this distribution *efficiently*, we also acknowledge that our method might not be practical. Since our work is purely theoretical, we leave practical implementation issues for future work, as discussed in Conclusion and Limitations.
>
> >What is $D_y$? It first appears in line 150 without definition and is later used in many places. I assume that  $D_y$ is intended to be the same as $Diam(\mathcal{Y})$ (or its upper bound) defined in line 113?
>
> Indeed, $D_y = Diam(\cal{Y})$. Thank you for catching this omission--we will add the definition to line 113.
>
> >...consider moving the hard instance around line 692 constructed for proving the lower bound to some place immediate after Assumptions 2.1 & 2.2 as a motivating example to illustrate these assumptions.
>
> Good suggestion. We will do this in the revision.
>
> >minor typos
>
> We will fix these. Thank you again for your careful reading of our manuscript.

---

> > ### Comment · Reviewer_aszX · 2025-08-06
> >
> > I want to thank the authors for their responses. All my questions are addressed, and I will keep my score.

---

### Official Review · Reviewer_aTVu · 2025-06-24

**Clarity:** 3
**Significance:** 2
**Originality:** 2
**Rating:** 4
**Confidence:** 3

**Summary:**

The authors study differentially private bilevel optimization with empirical loss and population loss. Their algorithm works even when the access to exact function evaluations is unavailable.

For convex outer objectives, the authors use exponential and regularized exponential mechanisms and establish upper and lower bounds on excess risk under both pure and approximate differential privacy, which is tight up to a $\mu_g$ term. For instance, the upper bound (Theorem 3.3) corresponds to
$$ \mathcal{O} \left( \Big( L_{f,x} + \frac{L_{f,y}\beta_{g,xy}}{\mu_g} + \frac{L_{g,y}\beta_{f,xy}}{\mu_g}  \Big) D_x \frac{\sqrt{d_x \log(1/\delta)}}{n \epsilon} \right) $$
And the lower bound (Theorem 3.9) corresponds to
$$ \Omega \left( \Big( L_{f,x} D_x + L_{f,y} D_y  \Big) D_x \frac{d_x}{n \epsilon} \right) $$

When the outer function is not convex, the authors develop algorithms that find approximate stationary points privately and show the following upper bound
$$ \mathcal{O} \left( K^{1/2} \Psi^{1/4} \beta_{\Psi}^{1/4} \frac{d_x^{1/2} \log^{1/4}(1/\delta)}{ (n\epsilon)^{3/4}}\right).$$
This result shows that the complexity is independent of the inner problem's dimension $d_y$.

**Questions:**

1. **Parameter dependency**: Theorems 3.2 and 3.3 involve $\mu_g$, while the lower bounds assume $\mu_g \approx L_{g,y}/D_y$. How does the lower bound extend to arbitrary values of $\mu_g$?

2. **Constraint assumptions**: Section 4 begins by assuming $\mathcal{X}$ is unconstrained, but Theorem 4.3 reverts to $\mathcal{X}$ being a convex subset. What is the scope of the unconstrained assumption?

3. **Optimality of non-convex rates**: The non-convex analysis achieves $1/(n\epsilon)^{3/4}$ accuracy, improving upon the standard $1/\sqrt{n\epsilon}$ rate in existing literature. Are there corresponding lower bounds establishing optimality for this problem class?

**Ethical Concerns:**

["NO or VERY MINOR ethics concerns only"]

**Final Justification:**

The authors answers clarify certain points, but not all of the problems are solved. I would keep the initial score.

**Limitations:**

yes

**Paper Formatting Concerns:**

No formatting issues found

**Quality:**

3

**Strengths And Weaknesses:**

The paper is well-structured with an accessible introduction that effectively contextualizes the challenges in differentially private bilevel optimization. The proposed algorithms and theoretical analysis are novel and technically sound. However, several aspects could benefit from greater clarity.

The paper would be strengthened by concrete bilevel optimization examples that illustrate why extending from convex to non-convex outer functions is practically important. Additionally, the treatment of problem parameters lacks precision—it's unclear whether constants like $\mu_g$ are inherent to the problem instance or chosen by the practitioner. The relationship between diameter bounds $D_x, D_y$ and the unconstrained setting introduced in Section 4 also requires clarification.

---

> ### Author Rebuttal · Authors · 2025-07-31
>
> Thank you for your thoughtful and detailed review. We respond to your comments below:
>
> >The paper would be strengthened by concrete bilevel optimization examples that illustrate why extending from convex to non-convex outer functions is practically important.
>
> There are **many important nonconvex BLO problems** that arise in practice: e.g., model selection and hyperparameter tuning, and some Stackelberg game models (see lines 128-129). Additionally, if the loss function $f$ is non-convex (e.g., as in deep learning), then the corresponding BLO problem is non-convex. We will elaborate on this  in the revision.
>
> >it's unclear whether constants like $\mu_g$ are inherent to the problem instance or chosen by the practitioner.
>
> $\mu_g$ parameter is **inherent to the problem instance**: it describes the strong convexity of function $g$. It is not chosen by the practitioner.
>
> >How does the lower bound extend to arbitrary values of $\mu_g$?
>
> Our lower bound construction satisfies $\mu_g \approx L_{g,y}/D_y$, or equivalently $D_y \approx L_{g,y}/\mu_g$. Thus, our lower bounds can equivalently be written in terms of $\mu_g$ by substituting the above relation for $D_y$. Our lower bound does not apply in the parameter regime $\mu_g \ll L_{g,y}/D_y$. Note that providing a lower bound in this regime is an open problem even for simple single-level DP ERM, as the lower bound construction of BST14 also requires $\mu \approx L/D$.
>
> >Section 4 begins by assuming $\mathcal{X}$ is unconstrained, but Theorem 4.3 reverts to $\mathcal{X}$ being a convex subset. What is the scope of the unconstrained assumption?
>
> All of **our results in Section 4 hold for both constrained and unconstrained settings** (see footnote 1). For Theorem 4.3, we additionally require the existence of a compact set $\mathcal{X}$ containing a global unconstrained minimizer. We will further clarify this in the revision.
>
> >Are there corresponding [tight] lower bounds establishing optimality for this [nonconvex] problem class?
>
> There are **no tight lower bounds for DP nonconvex optimization**, *even in the simpler single-level optimization setting*. We highlight this as an important problem for future work in the Conclusion.

---

> > ### Author Response · Authors · 2025-08-06
> >
> > Dear Reviewer aTVu,
> >
> > Thanks again for your review. Just a quick follow-up in case you haven’t had a chance to revisit our rebuttal. We aimed to directly address your concerns—particularly regarding the practical importance of (nonconvex) BLO, the role of $\mu_g$, and how our results in Section 4 hold for both constrained and unconstrained nonconvex BLO.
> >
> > If anything remains unclear, we’re happy to clarify. Otherwise, we’d be grateful if you’d consider whether the response warrants an updated score.
> >
> > Sincerely,
> >
> > Authors

---

> > ### Comment · Reviewer_aTVu · 2025-08-07
> >
> > I would like to thank the authors for the clarification. Consider the exchanges with other reviews and some of my questions (examples and lower bound) are not fully solved, I would like to keep the current score.

---

### Official Review · Reviewer_wGn3 · 2025-06-29

**Clarity:** 2
**Significance:** 3
**Originality:** 2
**Rating:** 4
**Confidence:** 2

**Summary:**

This paper addresses the problem of solving bilevel optimization (BLO) under differential privacy (DP), which is important for applications like hyperparameter tuning and meta-learning on sensitive data. The authors present novel upper and lower bounds for both convex and non-convex settings, covering empirical risk minimization (ERM) and stochastic optimization (SO). In the convex case, they match the rates of standard single-level DP optimization up to bilevel-specific complexity terms. In the non-convex setting, they design second-order DP algorithms with improved rates over previous works, including one that removes dependence on the inner problem’s dimension d_y. A significant contribution is the analysis of log-concave sampling under function evaluation errors, which enables efficient implementation of exponential mechanisms for BLO.

**Questions:**

1. In the definition of Bilevel ERM on page 2 following line 40, should the $\hat{y}_Z^\*$ inside function g() simply be $y$, given that the optimization is over $y$?
2. In section 1.1, the authors claim the algorithm lacks access to $\hat{y}_Z^\*$ in convex DP-BLO, but since the inner problem is strongly convex in $y$, a unique solution exists. Can the authors clarify why computing or approximating is not $\hat{y}_Z^\*$ feasible?
3. Could the authors explain more concretely why the warm-starting approach improves the upper bound? Also, since this approach uses the exponential mechanism followed by noisy gradient descent, how sensitive is the method’s performance to the choice of the warm-start point?
4. When the inner solution is computed non-privately, could this leak information about sensitive data to adversaries?
5. The grid-walk sampling approach is theoretically efficient, but how does it scale in high dimensions in practice? Are there benchmarks or estimates of runtime and sampling accuracy?

**Ethical Concerns:**

["NO or VERY MINOR ethics concerns only"]

**Final Justification:**

The complex sampling scheme is a drawback of the algorithm for practical implementation, which is not well solved or discussed in the paper. Given this, I think the paper is borderline, leaning toward acceptable.

**Limitations:**

Yes.

**Quality:**

2

**Strengths And Weaknesses:**

Strengths:
1. The paper tackles an important and underexplored problem: DP guarantees in BLO, which is crucial for privacy in hierarchical ML applications.
2. It provides both upper and lower bounds, contributing a nearly complete theoretical picture for convex BLO.
3. The proposed algorithms achieve state-of-the-art rates in the non-convex setting and offer dimension-independent bounds on d_y.
4. The paper introduces novel technical contributions, including new analysis for log-concave sampling with approximate function evaluations.
5. The theoretical results are comprehensive, covering both pure DP and approximate DP regimes, and highlighting separations between BLO and single-level DP optimization.

Weaknesses:
1. The work is entirely theoretical. No experimental validation is provided, which limits insight into practical performance and the real-world feasibility of the proposed methods.
2. The efficiency claims for the exponential and regularized exponential mechanisms rely on complex sampling schemes (e.g., grid-walk) that may be impractical at scale.

---

> ### Author Rebuttal · Authors · 2025-07-31
>
> Thank you for your thoughtful and detailed review. We respond to your comments below:
>
> >The work is entirely theoretical. No experimental validation is provided...
>
> We do not consider the lack of experiments in our paper to be a weakness: Like many accepted NeurIPS papers, our work is about understanding the fundamental complexity of an important problem, and we hope our paper will be judged on those terms. As discussed in the Conclusion and Limitations sections, empirical investigation is an important direction for future work on DP BLO.
>
> >The efficiency claims for the exponential and regularized exponential mechanisms rely on complex sampling schemes (e.g., grid-walk) that may be impractical at scale.
>
> We agree that our sampling algorithm may be complex to implement in large-scale practical applications, but do not consider this to be a weakness: The focus of our work is entirely theoretical–understanding the complexity bounds and developing polynomial-time algorithms that achieve these bounds. We leave it as future work to provide practical implementations of these algorithms.

---

> ### Comment · Reviewer_wGn3 · 2025-08-05
>
> Thank you for your reply, which helps clarify some of my concerns. Nevertheless, I still believe that, the complex sampling scheme would be a drawback of the algorithm for practical implementation, despite the theoretical side contribution of the work. Given that, I will maintain my original score.

---

### Official Review · Reviewer_pJMr · 2025-06-29

**Clarity:** 2
**Significance:** 3
**Originality:** 3
**Rating:** 3
**Confidence:** 4

**Summary:**

The paper studies the problem of DP bilevel optimizations for both convex and non-convex settings. For the case when the outer-level objective is convex, the authors provides both upper results for ERM and SCO cases. The authors also gives a matching lower bound result which show the tightness of their rates. For non-convex setting, they propose a new algorithm with improved rate over existing works.

**Questions:**

The questions are included in the section "Strengths and Weaknesses". I am willing to increase my rating if my concerns are addressed.

**Ethical Concerns:**

["NO or VERY MINOR ethics concerns only"]

**Limitations:**

yes

**Quality:**

2

**Strengths And Weaknesses:**

**Strengths**:

Bilevel optimization (BLO) is an important question with many applications, while the topic DP bilevel optimization literatures are quite underexplored. This paper deepens the understanding of DP BLO by providing improved rates and matching lower bound. Notably, the idea of using exponential mechanism to obtain rate independent of inner problem dimension is both novel and interesting.

**Weaknesses**:

1. The paper primarily focus on the setting where the outer objective $\Phi$ is convex. In practice, $\Phi$ is almost always non-convex, even in cases where the loss function $f$ is convex. The paper lacks the discussion or justification for when and why the outer objective $\Phi$ would be convex.
2. The proof of SCO result (second part of Theorem 3.3) appears to be either incorrect or at least unsound. The proof relies heavily on the results in [22] (Lemma B.7). However, the results in [22] only work in standard finite sum setting, where the empirical objective $\hat{\Phi}(x) = \frac{1}{n}\sum_{i\in [n]} f(x, z_i)$ for some loss function $f$. However, in BLO, the empirical objective $\hat{\Phi}(x) = \frac{1}{n}\sum_{i\in [n]} f(x, y^* (Z), z_i)$, where each individual function $f_i(\cdot)=f(\cdot, y^*(Z), z_i))$ depends on the entire dataset $Z$. This introduces correlations among $f_i(\cdot)$, $i\in [n]$ which will fundamentally affect the generalization analysis. As such, the generalization analysis needs to be carefully revisited in the context of BLO, rather than relying on results that apply to standard learning problems.
3. Section 3.2 is difficult to follow. My understanding is that it proposes an efficient implementation of the exponential mechanism introduced in Section 3.1. However, the discussion centers on newly defined functions $f(\theta)$ and $F(\theta)$, making it unclear how the results connect back to BLO notations $\hat{\Phi}$, $f$, and $g$. Can the result in Theorem 3.8 be directly applied to Theorems 3.1 or 3.3? If so, it would be helpful to explicitly state this, perhaps in the form of a corollary that illustrates how Theorem 3.8 leads to a concrete implementation in the context of BLO.
4. How large can the estimation error $\zeta(\theta)$ in Theorem 3.8 be while still achieving the stated bounds in Theorem 3.1? Furthermore, can this level of error be achieved under differential privacy constraints?

---

> ### Author Rebuttal · Authors · 2025-07-31
>
> Thank you for your thoughtful and detailed review. We respond to your comments below:
>
> >Discussion or justification for when and why the outer objective $\Phi$  would be convex.
>
> Convex BLO (i.e. convex  $\Phi$) is important, as evidenced by the large body of work studying this problem non-privately: see e.g., Ghadimi & Wang (2018), Ji & Liang (2023) and the references within. There are **many applications of convex BLO** problems, including: few-shot meta-learning with shared embedding model (Bertinetto et al., 2018), biased regularization in hyperparameter optimization (Grazzi et al., 2020), and fair resource allocation problem over communication networks (Srikant and Ying, 2013). With the additional space allowed for camera-ready, we will add a paragraph on these applications.
>
> >The proof of SCO result (second part of Theorem 3.3) appears to be either incorrect or at least unsound
>
> We thank the reviewer for this excellent and insightful comment. We agree that a rigorous generalization analysis is crucial, and we apologize for its omission in the initial draft.
>
> The reviewer is correct that applying a classic stability argument (e.g., via a "ghost sample") to the bilevel setting is non-trivial. The main difficulty arises from the need to bound the error propagation from the lower-level optimization. **During the rebuttal period, we used new sound analyses to obtain a generalization error bound of order $O(\min[1/n^{1/4}, \sqrt{(d_x+d_y)/n}\])$.**
>
> **Proof sketch:** As the first step, the $\mu_g$-strong convexity of the lower-level problem allows us to relate the solution error to the gradient of the true objective at the empirical minimizer: $|\hat{y}_Z^{\text{star}}(x) - y^{\text{star}}(x)\| \le \|\nabla_y G(x, \hat{y}_Z^{\text{star}}(x))\| / \mu_g$.
>
> From here, there are two ways  to bound the term on the right:
>
> 1. *Uniform Convergence:* A standard approach is to bound the gradient term using uniform concentration arguments. This is a sound method, but as the reviewer might expect, it results in a dimension-dependent generalization error of **$O(\sqrt((d_x+d_y)/n))$.**
> 2. *Variance Analysis:* To seek a dimension-independent bound, we analyzed the variance of $||\nabla_y G(x,\hat{y}_Z^{\text{star}}(x))||$. Our analysis utilizes the Efron-Stein Inequality. However  this path currently yields a worse generalization rate of **$O(1/n^{1/4})$.**
> In the revised manuscript, we will include the complete derivation. We will also add a discussion framing the gap between our generalization error bound and the optimal $O(n^{-½})$ bound as an important open problem in DP bilevel SCO.
>
> >Can the result in Theorem 3.8 be directly applied to Theorems 3.1 or 3.3?
>
> **Yes**, and we have updated the manuscript to make the direct application of Theorem 3.8 to our proposed mechanisms in Theorems 3.1 and 3.3 more explicit. The function $f(\theta)$ in Theorem 3.8 is instantiated as follows for our two algorithms:
> 1. For the **Exponential Mechanism** (Algorithm in (4)), we set $f(\theta) = (\epsilon/2s)\hat{\Phi}_Z(\theta) $ where $s$ is the sensitivity of the empirical objective.
> 2. For the **Regularized Exponential Mechanism** (Algorithm in (5)), we set $f(\theta)= k(\hat{\Phi}_Z(\theta)+ \mu \|\theta\|^2)$ where $k,\mu$ are two hyperparameters set correspondingly in Theorem B.8.
>
> >How large can the estimation error $\zeta(\theta)$  in Theorem 3.8 be while still achieving the stated bounds in Theorem 3.1? Furthermore, can this level of error be achieved under differential privacy constraints?
>
> It suffices to **set the estimation error $\zeta(\theta) = O(\varepsilon)$** in order to efficiently achieve DP and the desired excess risk bound. This follows from the distance bound in Theorem 3.8 and the following fact: if two non-negative random variables $X$ and $X′$ satisfy $\varepsilon$-differential privacy, then $E[X] \le e^{\varepsilon} E[X']$. Now take $X$ as the empirical risk under the conceptual perfect sampler and X' as the empirical risk the approximate sampler (i.e. actual implementation).

---

> > ### Comment · Reviewer_pJMr · 2025-08-03
> >
> > Thanks to the authors for the detailed response.
> >
> > On the convexity of the outer objective $\Phi$, my question was about the conditions under which we can guarantee that $\Phi$ is convex. The only example that comes to mind is the Moreau envelop construction that $\Phi(h) = \text{argmin}_w F(w) + \lambda \lVert w-h\rVert^2$ with convex $F(w)$. Can the authors provide any other examples or constructions such that the outer objective is convex?
> >
> > On the SCO results, the new bound is worse than the original one. Specifically, the revised bound scales either with the dimensions as $\sqrt{d_x + d_y}$ or with $1/n^{1/4}$, both of which seem potentially suboptimal. From my perspective, this affects the strength of the contributions to this paper. Furthermore, while the new bound seems technically correct to me, the extent of the changes suggests that additional reviews may be necessary.
> >
> > Given the above, I will maintain my original recommendation.

---

> ### Author Response · Authors · 2025-08-06
> **Significance of Convex $\Phi$ and SCO rates**
>
> We appreciate your engagement with our work, and we hope the clarifications below help convey why (convex) DP BLO represents a significant problem and one where this paper makes foundational progress.
>
> **Why convex $\Phi$ matters.**
> **The examples we cited in our rebuttal**—few-shot meta-learning with a shared embedding model (Bertinetto et al., 2018), biased regularization in hyperparameter optimization (Grazzi et al., 2020), and fair resource allocation in communication networks (Srikant and Ying, 2013)—**are indeed cases where $\Phi$ is convex**, as discussed in [1, p.7]. These problems arise in real-world applications and our algorithms are near-optimal.
>
> As another concrete problem class with convex $\Phi$, consider quadratic lower-level objectives of the form $g(x,y) = y^\top H y + x^\top J y + b^\top y + h(x)$
> where $H$ and $J$ are bounded Hessian and Jacobian matrices, and $f$ satisfies the assumptions in our paper. This also yields convex $\Phi$. We believe these examples illustrate that *convex BLO is not only mathematically tractable but also relevant to practical scenarios, making it a natural and important starting point for a rigorous DP theory.*
>
> **SCO rates and our overall contributions.**
> We acknowledge that the corrected SCO bound is weaker than the original statement, and we thank you for pointing out the earlier issue. While this bound is not yet optimal, it is the first established bound for DP bilevel SCO. Combined with our near-optimal ERM bounds (including the first lower bounds for DP BLO) and improved state-of-the-art rates for non-convex BLO, **we believe the overall theoretical contributions remain strong and impactful.**
>
> We hope these clarifications help convey both the significance of convex DP BLO and the breadth of our contributions, and we would be grateful if they inform your overall assessment of the paper.
>
>
> [1] Ji and Liang. *Lower Bounds and Accelerated Algorithms for Bilevel Optimization*, 2023.

---

### Note · Authors · 2025-08-13

We thank the reviewers and the Area Chair for their thoughtful engagement. We write these final remarks to emphasize the significance and novelty of our contributions and to consolidate the positive feedback from the discussion.

**Differentially Private Bilevel Optimization (DP BLO) is a fundamental and underexplored problem with growing importance** in hierarchical ML applications, including hyperparameter tuning, meta-learning, and fairness-aware optimization. Our submission offers what several reviewers recognized as foundational progress in this area---both for convex and non-convex settings.

**Key contributions include:**
1. The first **nearly matching upper and lower bounds** for convex DP BLO, establishing a **novel separation** from single-level optimization.
2. New algorithms with **improved $d_y$-independent rates in the non-convex setting**.
3. A **new analysis of log-concave sampling** under approximate evaluations, enabling efficient exponential mechanism implementations.

Multiple reviewers described our results as **“remarkable”** (aszX), **“nearly complete”** (wGn3), and **“novel and interesting”** (pJMr), and praised the paper’s **technical depth**, **clear writing**, and **breadth** across pure and approximate DP, ERM and SCO, and convex and non-convex regimes.

We also clarified misunderstandings during the discussion:
- We provided concrete examples where the outer objective $\Phi$ is convex in practical applications (e.g., few-shot meta-learning, fair resource allocation).
- We corrected the SCO bound, which remains, to our knowledge, the first for DP BLO—an inherently challenging setting that we will highlight as an open problem.

While the paper is theoretical, its scope and novelty align with the NeurIPS tradition of impactful theoretical work. We hope this submission will be evaluated in that light, and we thank the committee for considering our work.

---

### Decision · Program_Chairs · 2025-09-17

**Decision:**

Accept (poster)

**Comment:**

This paper addresses the problem of solving bilevel optimization (BLO) under differential privacy (DP) and give theoretical upper- and lower-bounds for the problem, establishing a novel separation from single-level optimization and give improved rates in the non-convex setting. The paper also contains a new analysis of log-concave sampling under approximate evaluations.
The reviewers all see the importance of the problem and find the paper well-motivated and all reviewers were appreciative of the theoretical analysis in this work. However, there was some disagreement regarding the strength of the convexity assumption and the applicability of the sampling scheme in this paper, flagging concerns regarding implementations of the work (not the lack of empirical validation, but questions such as which parameters a practitioner ought to know in advance).
Ultimately, we acknowledge that the paper presents a solid theoretical contribution to a problem which is of great importance to the ML community. We believe the merits of studying bilevel optimization with DP rigorously outweigh the negative review and therefore recommend acceptance.